# InterCode: Standardizing and Benchmarking Interactive Coding with Execution Feedback

**John Yang**  **Akshara Prabhakar**  **Karthik Narasimhan**  **Shunyu Yao**
Department of Computer Science, Princeton University
{jy1682, ap5697, karthikn, shunyuy}@princeton.edu

## Abstract

Humans write code in a fundamentally interactive manner and rely on constant execution feedback to correct errors, resolve ambiguities, and decompose tasks. While LLMs have recently exhibited promising coding capabilities, current coding benchmarks mostly consider a static instruction-to-code sequence transduction process, which has the potential for error propagation and a disconnect between the generated code and its final execution environment. To address this gap, we introduce InterCode, a lightweight, flexible, and easy-to-use framework of interactive coding as a standard reinforcement learning (RL) environment, with code as actions and execution feedback as observations. Our framework is language and platform agnostic, uses self-contained Docker environments to provide safe and reproducible execution, and is compatible out-of-the-box with traditional seq2seq coding methods, while enabling the development of new methods for interactive code generation. We use InterCode to create three interactive code environments with Bash, SQL, and Python as action spaces, leveraging data from the static NL2Bash [29], Spider [51], and MBPP [4] datasets. We demonstrate InterCode's viability as a testbed by evaluating multiple state-of-the-art LLMs configured with different prompting strategies such as ReAct [48] and Plan & Solve [40]. Our results showcase the benefits of interactive code generation and demonstrate that InterCode can serve as a challenging benchmark for advancing code understanding and generation capabilities. InterCode is designed to be easily extensible and can even be used to create new tasks such as Capture the Flag, a popular coding puzzle that is inherently multi-step and involves multiple programming languages. *

## 1 Introduction

The art of computer programming is naturally an interactive process. When a human programmer writes code, she relies on several iterations of a 'write-execute-test' loop in order to iteratively refine solutions, plan changes, test sub-modules, and solve ambiguities by checking execution behavior. While this is reminiscent of other human endeavors like writing, code compilation and execution produce exact results that provide a deterministic form of feedback to make the refinement process more straightforward. Depending on the observed results, programmers perform various levels of debugging and rewriting, and continue the process until their code satisfies the requirements.

There has been increasing interest in recent years around the development of models that can automatically generate code given a specification in natural language [17, 43, 13, 27, 24]. Powered by large-scale pre-training over thousands of codebases [2, 21, 18], these models have shown solid performance on static benchmarks like HumanEval [9], APPS [19], MBPP [4], CodeXGLUE [30]. However, generating code in a static, sequence-to-sequence or auto-regressive fashion has several drawbacks: 1) simple errors (even typos) can propagate and there is no chance for recovery or

---

*Code and data available at `https://intercode-benchmark.github.io/`

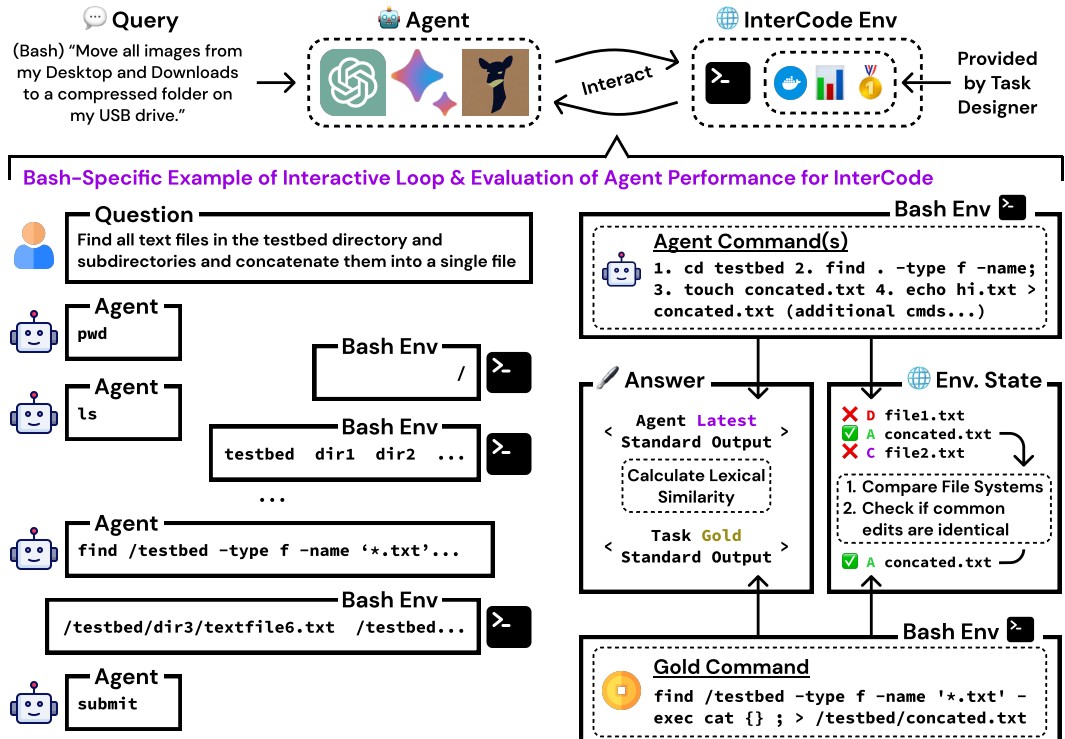

Figure 1: Overview of InterCode. Setting up an interactive code environment with InterCode requires a Dockerfile, dataset, reward function definition, and a small amount of subclass implementation. The interactive loop between agent and environment closely mirrors real world software development processes. While InterCode task performance is generally quantified as a binary 0/1 completion score, InterCode allows for the design of more complex evaluation criteria that can incorporate execution output and the effects of interaction on the state space.

revision, 2) there is a disconnect between the code generation process and its downstream execution on the desired software and hardware environment, and 3) there is little room for human intervention or collaboration in the code generation process.

Recently, some works have proposed the use of execution feedback or interaction [44] to benefit code generation models [23, 20, 45, 19]. However, these papers consider their own individual setup and are difficult to compare with one other due to the use of different compilers, execution environments, feedback signals, and assumptions on the interactive process such as human participation to create task descriptions or provide natural language feedback. This makes it difficult to compare existing methods for code generation and to clearly understand the benefits of interactive generation.

To address these issues, we propose InterCode, the first standard coding benchmark designed natively with an interactive execution environment. Closely mimicking the human decision-making process, InterCode allows a coding agent to interactively receive feedback from compilers/interpreters that execute its code, and to submit further refinements. We design InterCode to be like a standard reinforcement learning (RL) environment that requires minimal human intervention and one in which generated code is treated as actions, which are executed to reveal observations. Our framework is (1) language and platform agnostic and can easily be used for new coding problems, (2) uses self-contained Docker environments to provide safe execution, and (3) compatible out-of-the-box with traditional seq2seq generation methods, while also enabling and empowering the development of new interactive techniques.

We demonstrate the power of the framework by implementing Bash, SQL, and Python tasks within InterCode, building on pre-existing static datasets [58, 29, 4]. We perform experiments across diverse models and prompting methods, including ReAct [48] and Plan & Solve [40]. Our findings concretely showcase the benefits of interaction towards solving coding tasks, discuss the distribution of distinct code understanding challenges across different task settings, and explore the ease with which new tasks and datasets can be defined using InterCode.

To summarize, our paper makes the following contributions:

- We develop InterCode, a new, universal framework for interactive code generation, which provides ease of use, extensibility, and safety.
- Using InterCode, we perform a comprehensive evaluation of state-of-the-art models and identify several avenues for improvements.
- We release our framework as a new benchmark along with useful empirical tools to customize any new static code datasets into interactive tasks.

## 2 Related Work

**Interactive environments for coding.** Most coding benchmarks (e.g. SQL - Spider [51], KaggleD-BQA [25]; Bash - NLC2CMD [1], NL2Bash [29]; Python - HumanEval [9], APPS [19], MBPP [4], CodeXGLUE [30], CodeNet [35]) frame the coding problem as a sequence transduction problem (from instruction to code), rather than an interactive decision making problem with an execution environment. Attempts have been made to simulate interaction by developing conversational, dialogue-style [53, 52], multi-step problem solving [33] datasets, which involve pre-annotated human-designed queries. The work closest to InterCode has been recent explorations of Python Jupyter Notebooks as a natural choice for interactive coding [20, 23, 50]. However, task data and settings often constrain allowed actions to a closed domain of code and libraries [23, 50], use evaluation procedures or metrics that may not generalize [20], require human-in-the-loop participation (i.e. create task contexts, write problems, evaluate execution per task instance) [23], or are Python-exclusive [20, 23, 50, 45]. InterCode provides a more general purpose foundation defining interactive coding tasks that enables easy construction of diverse task settings, can have any programming language(s) as the action space, and has automatic, execution-based evaluation.

**Execution-based evaluation for coding.** Evaluation for NL-to-code generation models has recently shifted away from surface form similarity metrics (BLEU [34, 2], ROUGE [28], Exact Match) towards execution oriented ratings (unit tests [4, 9, 20, 23, 19], output matching [15, 20, 58]). The rigidity of surface form analysis overlooks code syntax features, ignores execution effect, or over-penalizes alternative solutions [59], On the contrary, execution-based assessment is a more thorough and comprehensive score of code functionality [19] and is a more natural fit for open-domain program usage that does not constrain code generation to a subset of the language space [45]. However, for newer benchmarks and datasets that put forth task definitions incorporating execution-based evaluation (APPS [19], ExeDS [20], ODEX [45]), the fundamental code generation task (Context + Code $\rightarrow$ Execution $\rightarrow$ Score) is still devoid of interaction. InterCode combines execution-based evaluation with flexible task construction, enabling more diverse problem-solving paradigms within a unified coding task formulation. InterCode's use of virtual containers as execution sandboxes protect against harmful actions and allow for advanced evaluation criteria beyond the aforementioned ones.

**Methods for interactive or execution-based coding.** The value of generative code models and interactive problem solving has motivated a recent proliferation of work to augment reasoning capabilities' of existing language models [48, 37, 40, 47, 56, 11] or propose new modeling techniques to tackle coding as a sequential decision making and reasoning tasks [6, 10, 16, 27, 8, 24], of which evaluation is unit test based. Approaches that leverage execution typically use re-ranking [57, 32, 49, 54] or majority vote [10, 27, 36] to decide on a final prediction. Additional work also explores incorporating human-in-the-loop [7, 22], compilers [41], and text [42, 55] feedback. A common thread among these contributions is that 1) the task setting can only provide the investigated form of feedback and 2) sought-after capabilities are exemplified by strong performance on favorably curated tasks and datasets, rendering comparisons across benchmarks tedious. InterCode has the potential to standardize the evaluation of these methods because 1) the interactive coding task is a conglomeration of many interesting interaction, reasoning, and decision-making challenges and 2) InterCode's task construction makes it possible to incorporate a wide variety of sources of feedback.

## 3 The InterCode Benchmark

### 3.1 Formulation

The InterCode benchmark formalizes interactive coding with execution feedback as a partially observable Markov decision process (POMDP) $(\mathcal{U}, \mathcal{S}, \mathcal{A}, \mathcal{O}, \mathcal{T}, \mathcal{R})$ with instruction space $\mathcal{U}$, state

| Action Space | Environment 🐳 | Dataset 📊 | Reward Function 🥇 |
|---|---|---|---|
| Bash | Ubuntu Terminal | NL2Bash [29] (200) | Latest Std. Output + File System $\Delta$ |
| SQL | MySQL Database | Spider 1.0 [51] (1034) | Latest Std. Output |
| Python | Python Interpreter | MBPP [4] (117) | Submitted Function |

Table 1: Rundown of the two environments with Bash and SQL as action spaces developed using the InterCode framework. The numbers in parentheses refer to the number of task instances adopted from each dataset. Each environment is defined in under 200 lines of code total. Specific discussion of the environment construction and reward function can be found in § A.2 and § A.3

space $\mathcal{S}$, action space $\mathcal{A}$, observation space $\mathcal{O}$, transition function $\mathcal{T} : \mathcal{S} \times \mathcal{A} \rightarrow \mathcal{S}$, and reward function $\mathcal{R} : \mathcal{S} \times \mathcal{A} \rightarrow [0, 1]$. Given a coding instruction $u \in \mathcal{U}$ in natural language, an agent issues code or a special submit keyword as an action $a_t \in \mathcal{A}$. An action is *admissible* [46] if it can be parsed and executed in the compiler/interpreter environment, and an admissible action incurs a change in the latent state space $s_{t+1} \in \mathcal{S}$, and an execution feedback as observation $o_{t+1} \in \mathcal{O}$. The interaction loop repeats until the submit action is issued, wherein the task episode ends and a reward $r = \mathcal{R}(s_T, \text{submit}) \in [0, 1]$ is computed, with 1 representing task completion. We use the **Success Rate (SR)** metric, defined as the proportion of task episodes where $r = 1$. We also define the **Error %** metric, which is the percentage of *non* admissible actions across task episodes.

### 3.2 Construction pipeline

At a high level, InterCode decomposes the construction of an interactive coding task into three **modular** parts: (1) environment construction, (2) data collection, and (3) reward design. This workflow allows for the safe execution of transition functions, flexible reward design, and convenient adaptation of existing instructions to an interactive setting.

**Docker-based environments.** InterCode uses Docker [31] virtual containers as a general-purpose execution sandbox. Given a Dockerfile that defines a system and execution entrypoint, InterCode creates a corresponding, stateful virtual container that hosts the desired state space and transition function. We choose Docker as the basis of InterCode's environment construction for its safe execution in virtual containers, reproducibility of a Dockerfile across any Docker-equipped machine, and excellent coverage of application code, libraries, and dependencies offered by the Dockerfile DSL.

**Data collection.** InterCode requires that a dataset has at minimum two fields: query, a natural language instruction $u \in \mathcal{U}$, and gold, an answer or code block that is a procedure for generating the correct answer. We define these conditions to make it easy to adapt existing text-to-code datasets to an interactive setting while also leaving plenty of bandwidth for constructing new tasks and datasets.

**Reward design.** Across a single task episode, the action, observation, and state modification (if any) per interaction loop are implicitly logged by InterCode. InterCode's default reward function determines task completion via an exact match of the agent's execution output (observation and state modifications) against the gold command, where 1 is awarded only if all components match. Since Exact Match is usually too stringent of an evaluation criteria, InterCode exposes a reward function endpoint that has access to both the interaction history and the execution container, allowing for custom reward function definitions that can incorporate multiple signals.

### 3.3 Implementations

Following the procedure discussed in Section 3.2, we create two separate InterCode based environments where Bash and SQL are the action spaces respectively. Table 1 summarizes them.

**InterCode-Bash.** We define a bash shell within an Ubuntu Operating System as the task setting. To evaluate an agent's ability to adapt generations to different situations, we architect four distinct file systems that can be swapped into the Bash environment by changing a single line in the Dockerfile.

We bootstrap the NL2Bash [29] dataset (which lacks specificity in queries and grounding to any underlying file system, preventing it from being used directly for interactive evaluations) to create an interactive coding task where an agent completes an instruction via bash actions. Transferring NL2Bash to the interactive task setting requires simple transformations to ground instructions and gold code blocks in the file system. First, we consider a subset of 1000 commands with each

having $\geq 4$ utilities. We then filter out commands that are non-UNIX, non-Linux, or use utilities we currently do not support (eg. "ssh", "sudo", time, and GUI-dependent utilities). Finally, we enhance under-specified commands with specific file names/directory names/paths and update deprecated utilities/flags. The resulting 200 commands are grouped into 4 disjoint sets, 3 of which were grounded to custom-designed file systems, while one set is file-system agnostic. This categorization allows for a comprehensive evaluation of different command-grounding scenarios.

The InterCode-Bash dataset instructions typically make one or both of the following two types of requests. It either 1. Requests information that can be answered via execution output (i.e. "How many files...", "What is the size of...", "Where is `<file>` stored?") or 2. Requests a change to the location/-configuration/content of a file or folder (i.e. "Move `dir1` folder...", "Set permissions of...", "Append a line to..."). Therefore, we define a custom reward function that evaluates an agent's performance against file system modifications and the latest execution output. Execution output is graded with a simple lexical similarity function. File system assessment is done in two parts. First, a comparison of the agent's and `gold` command's list of file system changes (list of [`path, modification type` $\in$ [`added, changed, deleted`]] entries) reveals any extraneous or missing changes. Second, `md5sum` hashes of each commonly edited file path are compared to determine if an added or changed file was altered correctly. A max score of 1 is achieved only if the correct file paths are changed, the changes are correct, and the latest execution output matches the gold command output exactly. Additional Bash statistics and design details are discussed in § A.2.

**InterCode-SQL.** We write a `Dockerfile` that defines a SQL interpreter within a MySQL database as the task setting. To create the databases and tables necessary for the task dataset, we write type resolution scripts and perform database conversions using the `sqlite3mysql` [38] Python library to adapt the Spider [51] database and table schema to a MySQL format. We then consolidate all setup code into a single, unified MySQL `.sql` dump that contains the complete set of schemas for all tables across 20 different databases. On container start-up, this file is invoked automatically, creating and populating databases with tables and tables with records.

The Spider [51] dataset is a large-scale cross-domain dataset originally meant for evaluating SQL query generations from natural language questions. We adapt the development set, which contains 1034 task instances, and remove all extraneous columns aside from the natural language questions and gold SQL command. The `instruction` and `gold` values do not require any additional pre-processing to be compatible with the MySQL task environment.

Finally, we employ Intersection over Union (*IoU*), or more formally the Jaccard Index, to quantify the correctness of the latest execution output generated by the agent against the gold output, where both outputs are a list of records. A non-tabular execution output receives a reward of 0 by default. Among the items that lie in the intersection of the agent and gold execution outputs, we also apply a penalty if the records are in the incorrect order. To quantify how sorted the agent output is relative to the gold output, we lean on Kendall's $\tau$ and adjust the output range to $[0, 1]$. The *IoU* score is then directly scaled by this coefficient. All in all, only a correctly ordered list with the exact set of records found in the gold output receives a score of 1. Visualizations like Figure 1 for SQL along with a more extensive implementation discussion for this environment are in § A.3

**InterCode-Python.** In this setting, we define a Python interpreter running within an Ubuntu operating System as the task setting. The Dockerfile can be configured to run any Python version. The interpreter is not initialized with any dependencies, but PyPI packages can be installed and used by the agent.

We use the MBPP [4] dataset which presents the code completion task of synthesizing Python code from a method header and docstring. Evaluation of correctness is performed with an associated set of unit tests given by MBPP. The MBPP dataset is straightforward to adapt to the interactive setting, requiring no modifications to the query or evaluation components. Finally, we directly inherit MBPP's evaluation procedure of proportion of unit tests passed. With InterCode, it is easy to use existing datasets to evaluate how well models can use different programming languages as actions.

**Validations.** To verify the functionality of action execution in the task environment and the correctness of custom reward functions, we write testing scripts for both Bash and SQL that pass the gold command in as a dummy agent's action to ensure that the command is admissible and executes without error, and to verify that the reward received by the command is 1. To confirm that InterCode's dataset specification is enforced across multiple accepted file formats, we define a custom InterCode data loader class which is then rigorously unit tested.

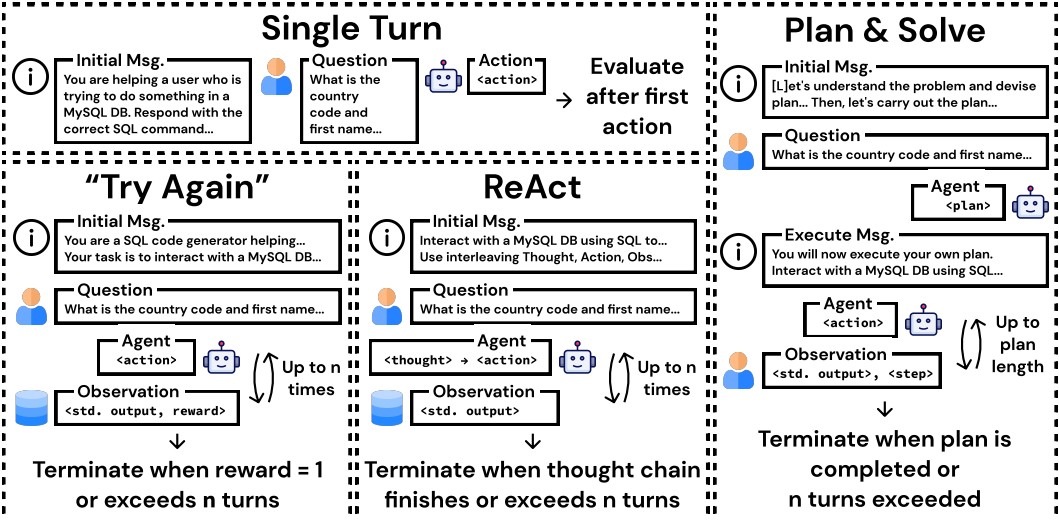

Figure 2: Overview of Prompting Strategies adjusted for evaluation on InterCode. The "Try Again" termination constraint is conditioned on reward = 1, while ReAct [48] and Plan & Solve [40] are determined by the agent itself. This is because the purpose of the "Try Again" method is to explore how capable agents are at error correction from feedback, while the other two are more concerned with the overall success of general problem-solving strategies.

## 4  Methods

We perform preliminary experiments to gauge the proficiency and behavior of current large language models on interactive coding tasks with Bash and SQL. To observe and elicit relevant reasoning skills, we draw on several existing prompting strategies that have been put forth to augment language models' reasoning and problem-solving skills. We apply these prompting strategies to models across the following three families: OpenAI (`text-davinci-003`, `gpt-3.5-turbo`, `gpt-4`), PaLM-2 (`text-bison-001`, `chat-bison-001`) [3], and Open Source (Vicuna-13B [12], StarChat-16B [26]).

Figure 2 visualizes the four adjusted prompting strategies we evaluate on InterCode.

**Single Turn** is a zero-shot attempt. A model is given a simple description of the task setting and asked to generate code in a specific programming language that would address the query. The first generation in response to the user's question is then evaluated in the InterCode environment.

**"Try Again"** is an iterative feedback set up. In the initial message, the agent is informed of the task setting and its interactive nature; an agent has multiple turns to interact with the system, wherein each turn, upon generating an action, the execution output of the action is fed back as an observation. This continues until a reward of 1 (task completion) is achieved or the number of turns ($n$) is exhausted. The agent's position in this approach is meant to mirror human software development as closely as possible. The goal of this method is to probe language models' raw interactive coding abilities in addition to illustrating the benefits and different challenges that arise in interactive coding tasks.

**ReAct and Plan & Solve.** We write prompts and design workflows that follow the text and task configurations described in ReAct [48] and Plan & Solve [40] as faithfully as possible. For these two approaches, the termination of a task episode is conditioned upon the agent's own judgment, as our goal with these methods is to gauge the transferability to and efficacy of existing reasoning frameworks with respect to the interactive coding task. Full prompt templates are included in §B.7.

## 5  Experiments

### 5.1  Base models comparison

**Task performances.** We first compare the success rate of models in the Single Turn and Try Again settings for both the InterCode-Bash and SQL datasets. From Table 2 and Table 3, we observe

| InterCode-SQL | Single Turn | | | | | Try Again ($n = 10$) | | | | |
|---|---|---|---|---|---|---|---|---|---|---|
| Model / Hardness | Easy | Med | Hard | Extra | All | Easy | Med | Hard | Extra | All |
| text-davinci-003 | 20.6 | 4.9 | 1.7 | 0.0 | 7.4 | 32.4 | 14.6 | 5.2 | 4.2 | 15.6 |
| gpt-3.5-turbo | 22.6 | 8.3 | **5.7** | **3.6** | 10.5 | 72.5 | 44.3 | 43.7 | 21.1 | 47.3 |
| gpt-4 | 19.8 | 7.2 | 4.6 | 3.0 | 9.1 | **87.5** | **76.7** | **66.7** | **52.4** | **73.7** |
| text-bison-001 | **23.8** | **10.9** | **5.7** | 0.6 | **11.5** | 27.0 | 12.3 | 5.7 | 0.6 | 12.9 |
| chat-bison-001 | 18.5 | 6.5 | 4.0 | 0.0 | 7.9 | 22.2 | 7.8 | 6.9 | 0.0 | 9.9 |
| Vicuna-13B | 8.1 | 1.3 | 0.6 | 0.0 | 2.6 | 18.9 | 3.4 | 1.7 | 0.0 | 6.3 |
| StarChat-16B | 21.8 | 7.4 | 2.9 | 0.0 | 8.9 | 22.3 | 8.5 | 2.9 | 1.2 | 9.7 |

Table 2: Success Rate for single vs. multi turn evaluation on InterCode-SQL (refer §A.3). Query difficulty is adopted from Spider [51]. Best metrics are in **bold**.

| InterCode-Bash | Single Turn | | | | | Try Again ($n = 10$) | | | | |
|---|---|---|---|---|---|---|---|---|---|---|
| Model / File System | 1 | 2 | 3 | 4 | All | 1 | 2 | 3 | 4 | All |
| text-davinci-003 | 10.0 | 32.1 | 28.8 | 33.3 | 24.6 | 30.0 | **52.8** | 32.2 | 44.4 | 38.7 |
| gpt-3.5-turbo | **30.0** | **39.6** | 33.3 | 37.0 | **34.5** | **45.0** | 49.1 | 45.0 | 48.1 | 46.5 |
| gpt-4 | 25.0 | 37.7 | **36.7** | **40.7** | 34.0 | 41.7 | 47.2 | **51.7** | **59.2** | **48.5** |
| text-bison-001 | 15.0 | 22.6 | 11.7 | 22.2 | 17.0 | 23.3 | 28.3 | 16.7 | 22.2 | 22.5 |
| chat-bison-001 | 12.1 | 22.5 | 16.7 | 22.2 | 17.7 | 13.8 | 24.5 | 18.3 | 22.2 | 19.2 |
| Vicuna-13B | 10.0 | 24.5 | 18.3 | 7.4 | 16.0 | 15.0 | 35.8 | 25.0 | 22.2 | 24.5 |
| StarChat-16B | 15.5 | 22.6 | 13.3 | 22.2 | 17.7 | 17.2 | 30.2 | 21.7 | 29.6 | 23.7 |

Table 3: Success Rate across file systems for single vs. multi-turn evaluation on InterCode-Bash (refer §A.2). To evaluate models' ability to interact with different task settings, we evaluate disjoint sets of Bash instructions across four different file systems. Best metrics are in **bold**.

that performance across different levels of task difficulty (SQL) and different file systems (Bash) is superior in the interactive setting for all models, with a notable multi-fold increase for GPT-4 ($9.1\% \rightarrow 73.7\%$) on the InterCode-SQL task.

**Analysis of interactions.** Manual inspection of trajectory logs indicates that models actively exercise later turns for discovering relevant context, correcting errors via execution feedback as observations, and solving problems via iteratively constructing and editing actions as affirmed by Figure 3. In addition, models also demonstrate a level of planning and modular problem solving; for instructions with `gold` commands that chain multiple commands together (i.e. with |, >, or ; in `bash`) or consist of multiple sub-problems (i.e. subqueries in `SQL`), models will use observations from solving smaller sub-problems in earlier turns to compose the higher-order action. Trajectories that exhibit these phenomena are in § B.4

**Failure cases.** With that said, both Figure 3 exhibits a plateauing in Success Rate and and Error %. This suggests that as the amount of context and feedback builds up, models are less capable of discerning relevant past history toward future actions. In late-turn scenarios, task episode trajectories often reveal repetition of earlier actions, a failure to effectively use recent observations towards deciding an appropriate next action, or an inability to recognize that a current problem-solving chain of thought is inconclusive or futile. This is particularly evident for `hard` and `extra` level InterCode-SQL task instructions that require context spanning across several tables and actions that incorporate multiple clauses. We note that even when the full schema of all tables and their descriptions are offered in addition to the original instructions, models still benefit greatly from using interaction to experiment with different `JOIN` and filtering operators across multiple turns, as demonstrated in § B.2. A larger context window size, retrieval of useful memory, and more adaptive reasoning paradigms are just a handful of potential solutions to overcoming such challenges.

## 5.2 Prompting strategy comparison

Initiating language agents with prompting strategies that encourage different forms of reasoning toward problem-solving improves performance on the interactive coding task to varying degrees. Table 4 presents side-by-side comparisons of the success rate, number of turns, and error rate per strategy. Compared to Try Again, which lacks specific guidance on leveraging multiple turns, more

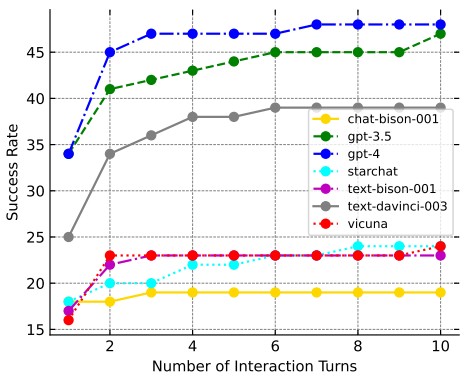

(a) Success rate vs. turns for InterCode-Bash

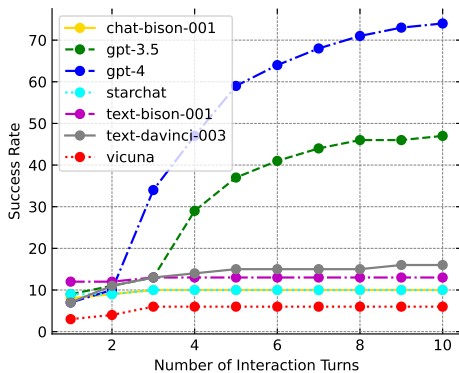

(b) Success rate vs. turns for InterCode-SQL

Figure 3: Growth in Success Rate with increase in number of interaction turns across models configured with Try Again prompting strategy for InterCode-Bash and SQL tasks.

| | Try Again ($n = 10$) | | | ReAct ($n = 10$) | | | Plan & Solve | | |
|---|---|---|---|---|---|---|---|---|---|
| | SR | Turns | Error % | SR | Turns | Error % | SR | Turns | Error % |
| SQL | 47.3 | 7.25 | 46.4 | **58.7** | 5.30 | **6.94** | 49.1 | **4.29** | 16.2 |
| Bash | **46.5** | 6.15 | 24.9 | 20.5 | **4.40** | **20.4** | 28.0 | 6.65 | 53.3 |

Table 4: Comparison of different prompting strategies across the entire InterCode-SQL and InterCode-Bash datasets using `gpt-3.5-turbo` as the base model. *Turns* refers to the average number of turns taken for a single task episode. For Try Again and ReAct, the max number of turns $n = 10$. The highest Success Rate, fewest Turns, and lowest Error % are highlighted per dataset since they reflect more accuracy and efficient task solving. Best metrics are in **bold**.

explicit reasoning frameworks such as ReAct and Plan & Solve policies generally achieve higher success rates (SQL: $47.3\% \rightarrow 58.7\%$) with fewer turns and a higher rate of admissible commands.

**Different tasks present different learning challenges.** An important skill to solving the InterCode-SQL task is the ability to discover context and construct actions conditionally based on information revealed in prior observations. Given that InterCode-SQL task instructions are phrased most commonly as questions, adapting to the task setting and new information discovered along the way puts more emphasis on error correction and context discovery. On the other hand, the more declarative and multi-step nature of the InterCode-Bash task instructions is more aptly solved by planning and modular task completion. These distinctions manifest in the Plan & Solve strategy's performance gap between the InterCode-SQL and InterCode-Bash tasks; while Plan & Solve encourages a model to decompose problems into more manageable steps, the strategy is less favorable towards adjusting on the fly in response to execution feedback. Example trajectories supporting these claims are in § B.4.

**More adaptive reasoning is favorable.** Compared to "imperative" reasoning paradigms such as Plan & Solve which prescribe a relatively rigid procedure, more flexible frameworks like ReAct, which do not enforce any particular logical formula or roadmap, are more conducive to eliciting a broader set of reasoning capabilities. However, while ReAct's performance is generally superior to Plan & Solve, tasks solved by *both* strategies with `gpt-3.5-turbo` make up $57\%$ ($407/708$) and $27.6\%$ ($21/76$) of the union of all successfully solved InterCode-SQL and InterCode-Bash tasks respectively. This discrepancy highlights a trade-off between the guidance and structural constraints that are inherent to prompting strategies; schemes that draw out specific reasoning patterns often overlook other equally useful capabilities. InterCode's interactive coding task can serve as a strong litmus test toward more adaptable, variegated model reasoning.

### 5.3  New tasks & datasets opportunities

InterCode's task formulation, modular design, flexible task construction, and use of virtual containers enable task designers to manifest new, complex, code-driven tasks, where completion is much more

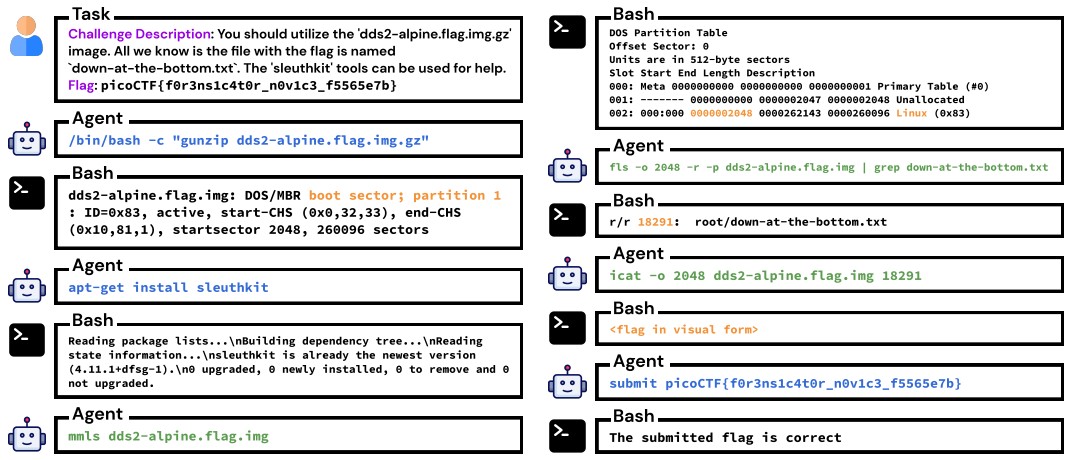

Figure 4: GPT-4's interaction trajectory for a binary exploitation CTF task. This requires proficiency in Bash and Python, among additional knowledge and reasoning. Orange text and arrows highlight the feedback that the model attends to in generating the next action. In last step, agent submits flag.

attainable through interaction. We draw inspiration from Capture the Flag (CTF) [14], a competitive cybersecurity game that requires expertise in coding, cryptography (i.e. binary exploitation, forensics), reverse engineering, and recognizing security vulnerabilities to accomplish the primary objective of discovering encrypted "flags" concealed within code snippets or file systems. Compared to InterCode-Bash & -SQL, CTF is much more complicated, requiring an agent to exercise knowledge of multiple coding languages, modularize a higher-order objective into sub-problems, construct multi-step plans towards solving each problem, and adjust strategy when a plan fails to yield any useful insights.

We establish InterCode-CTF, a new dataset consisting of 100 CTF objectives from picoCTF [39]. Following the interactive coding task formulation, each task instance in InterCode-CTF is given as a `<instruction, assets, hidden flag>` tuple. We first construct a Bourne Shell within an Ubuntu OS as the task environment. Here, InterCode's use of virtual containers is crucial, as necessary actions can be irreversibly damaging on real systems (i.e. `rm -rf`, `sudo` access). Per task instance, the associated assets (e.g., images, executables, code), necessary for task completion, are copied into the OS file system. Given this setting, a task worker must understand the given material and investigate the assets to develop potential solutions. Executing a successful approach must be done across multiple steps with various conditionals, where the execution feedback of a prior step could have a significant effect on the next step. Figure 4 spotlights the diverse skills needed for CTF.

# 6 Discussion

**Conclusion.** We have developed InterCode, a novel lightweight framework that facilitates interaction between Language Models and the underlying environment, enabling them to mimic the human approach to language-to-code generation. Our framework has shown promising results when applied to state-of-the-art models using different prompting styles. It effectively leverages the capabilities of LMs to break down complex tasks and recover from errors within a secure and isolated environment. The ability to seamlessly convert existing datasets into the interactive format using `InterCodeEnv` API, and furthermore, the Bash and SQL environments, empowers task designers to construct new tasks to unlock the plethora of challenges that await in the space of interactive coding.

**Limitations and future directions.** We point out several current limitations of InterCode. At this time, the number of InterCode based environments is limited to Bash, SQL, and Python action spaces and datasets; within the near future, we plan to expand the number of offerings to cover a wider set of programming languages and datasets that should further deliver on InterCode's purported promises of efficient and expressive task construction. Second, the CTF dataset is limited to just four task instances due to our manual curation procedure. We hope to release more formal work soon that provides a more thorough analysis of the reasoning and collaboration challenges of the CTF task along with a more extensive dataset for evaluation purposes.

## Acknowledgements

We thank Xiao Liu for the Vicuna/Alpaca APIs, Carlos Jimenez and Yuhan Liu for trying our code, and Princeton NLP Group for helpful discussion and feedback in general. We acknowledge support from the National Science Foundation under Grant No. 2107048. Any opinions, findings, and conclusions or recommendations expressed in this material are those of the author(s) and do not necessarily reflect the views of the National Science Foundation.

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

# Appendix

In this appendix, we provide additional details about the implementation and usage of the InterCode framework and the `InterCodeEnv` interface. We also provide visualizations and analyses of additional experiments to demonstrate InterCode's utility and garner further insight into the extent of current models' performance on the interactive coding task. The full template for each prompting strategy is also included. Finally, we also discuss some of the impacts, risks, and limitations of our work. The webpage for InterCode is `https://intercode-benchmark.github.io/`. The code for InterCode is `https://github.com/princeton-nlp/intercode`; the link is also included on the InterCode webpage.

## A  Environment Details

### A.1  InterCode Interface

The InterCode interface inherits the OpenAI gym [5] environment API definition. Specifically, `InterCodeEnv` is written as an abstract class that primarily handles the main execution logic for processing code interactions, in addition to logging, data management, and sand-boxed execution, along with both environment-level and task-level customization.

`InterCodeEnv` exposes the following API. Creating an interactive coding environment requires defining a subclass of `InterCodeEnv`. The methods denoted with an asterisk can be overridden for the purposes of customization.

`__init__(self, data_path: str, image_name: str, **kwargs)`

- Validates that the dataset specified by `data_path` is formatted correctly and can be used in an interactive setting.
- Uses the Docker image specified by `image_name` to create and connect with a Docker container instance of the image.
- Initializes Logging Handler
- Keyword arguments:
    - `verbose (bool)`: If true, logging is enabled and environment interactions are shown to standard output
    - `traj_dir (str)`: If a valid path is provided, task episode summaries are saved to the given directory (generated by `save_trajectory`)
    - `preprocess (callable)`: If provided, this function is run before every task episode. It is a way to provide task instance-specific customization of the execution environment.

`reset(self, index: int = None) -> Tuple[str, Dict]`

- Retrieves task record from data loader
- Calls `reset_container`
- Reset task level logger, instance variables

`step(self, action: str) -> Tuple[str, int, bool, Dict]`

- Log (action, observation)
- Invoke `exec_action` on action argument
- If `action=submit`, invoke `get_reward`, `save_trajectory`

`save_trajectory(self)`

- Saves task metadata, (action, obs.) sequence, and reward info to `.json` in `traj_dir`

`close(self)`

- Safely exit or stop any resources (i.e. docker container) used by the environment

`* execute_action(self, action: str)`

- Defines how the action is executed within the context of the docker container.
- Requires impl. because the Dockerfile definition, particularly its *entrypoint*, affects how an action would be invoked within the container.

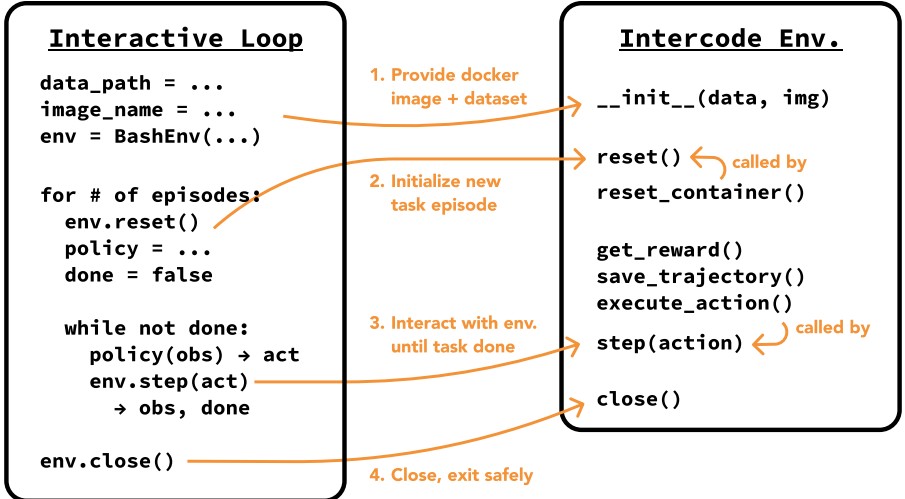

Figure 5: Visualization demonstrating the intended invocations and usage of the `InterCodeEnv` interface, along with how the functions requiring implementation (`get_reward()`, `execute_action()`, `reset_container()`) are called by the methods of the main interactive loop.

- Default impl. passes the `action` string directly into a `self.container.exec(action)` call, which invokes the action in the environment and returns execution output. A timeout is imposed on execution duration.

* `get_reward(self) -> Tuple[float, Dict]`

  - Handles reward calculation of actions with respect to the gold command(s) for a task episode.
  - Requires impl. because the concept and scoring for task completion varies across datasets and environments.

* `reset_container(self)`

  - Handles resetting of execution container (i.e. resetting file system to original state).
  - Requires impl. because the approach to restoring a setting to its initial state varies.

Figure 5 conveys how each of these methods are invoked and how they related to one another. In summary, the technicalities for setting up an interactive coding task for a specific system with one or more programming languages as the action space involve:

- Defining a `Dockerfile`
- Providing a dataset with the `query` and `gold` fields
- (Optional) Defining a reward (`get_reward`) function to define task completion.
- (Optional) Creating an `InterCodeEnv` subclass that overrides the `execute_action` and `get_reward` methods

## A.2 Bash Environment

**Environment Definition.** The `Dockerfile` defining the Bash-based environment is founded on the LTS version of the Ubuntu operating system. Several Linux dependencies that can potentially be used by an agent to address instructions in the InterCode-Bash Dataset are then installed via the Advanced Package Tool (`apt`) interface. Next, a shell script is invoked within the `Dockerfile` to initialize one of the three file systems displayed in Figure 6. The shell script consists of a simple sequence of `mkdir`, `touch`, and `echo` commands to deterministically create and populate the content of multiple files and folders. Finally, `git` is configured for the purposes of determining file diffs per task episode (`git status -s`) and resetting an environment to its original state (`git reset -hard; git clean -fd;`) before the beginning of a new task episode. The original code for the `Dockerfile` along with the file system creation scripts can be found on the project GitHub repository.

**Dataset Details.** The log-frequency distribution of the top-50 utilities is displayed in Figure 7. The NL2Bash [29] dataset is made available for use under the GPLv3 License. To assess the

generalizability of our approach, we designed three distinct file systems to accommodate the bash commands we collected. A key consideration during the construction of these file systems was to ensure that a significant portion of the executed commands would not result in operations that yield no changes. This deliberate design choice aimed to provide a more comprehensive assessment of our approach's adaptability and effectiveness across various scenarios and command executions. The file systems encompass a wide range of file types, including text files (.txt), program files (.c, .java, .py), compressed files (.gz), shell scripts (.sh), PHP scripts (.php), JSON files (.json), documents (.doc), spreadsheets (.csv), webpages (.html), database schemas (.sql), hidden files, and files with special characters in their names, convoluted folder hierarchies. Their directory structures are illustrated in Figure 6. For simplicity, we consider the top-level folder created within the root directory (`testbed`, `system`, `workspace`) as the root of each file system. This root folder contains files and sub-folders that necessitate access and manipulation, while changes are monitored throughout the entire container to accurately evaluate the models' actions. Notably, we intentionally designed file system 1 to be more intricate and encompass relatively challenging bash tasks compared to the other two file systems. Thereby, the models' performance is relatively lower for file system 1.

**Reward Function.** Evaluation of an agent's trajectory across a single task episode towards carrying out the given instruction is determined by modifications to the file system and the latest execution output. The instructions found in the InterCode-Bash dataset fall under one of two buckets: it either 1. Requests information about the file system that can be answered via execution output generated from a correct sequence of Bash actions (i.e. "How many files...", "What is the size of...", "Where is the .png image stored?") or 2. Requests a change to the location, configuration, or content of a file or folder (i.e. "Move the `dir1` folder from...", "Set the permissions to...", "Append a line to..."). Any relevant correct changes are therefore captured by considering both execution output and file system modifications during evaluation.

We define $A$ and $G$ as the outputs of the `agent` and `gold` commands respectively, where $A_{out}$ and $G_{out}$ refer to the execution output, and $A_{fs}$ and $G_{fs}$ refer to a list of entries reflecting file system modifications, where each entry is `[file path, modification type ∈ [added, changed, deleted]]`. We then formally define the reward function as follows:

$$\mathcal{R} = 0.34 * \texttt{similarity}(A_{out}, G_{out})$$
$$+0.33 * (1 - \texttt{erf}(|A_{fs} \cup G_{fs} - A_{fs} \cap G_{fs}|)) +$$
$$+0.33 * \frac{\texttt{is\_correct}(A_{fs} \cap G_{fs})}{A_{fs} \cap G_{fs}} \tag{1}$$

Where `similarity` refers to lexical similarity, which is determined by the cosine similarity score between TF-IDF vectors (calculated with `TfidfVectorizer` from `scikit-learn`) of the two execution outputs. The second component of the reward function reflects the number of file system modifications that were either not completed or not necessary; the error associated with the total number of misses is constrained to the range `[0,1]` using the Gauss error function (`erf`), where $0$ corresponds to no file system modification mistakes. The third component checks what proportion of paths altered by both `agent` and `gold` were modified correctly. The `is_correct` function returns the number of file paths that were changed correctly, determined by checking whether the `md5sum` hashes of each file path are identical for `agent` and `gold`. If $A_{fs} \cap G_{fs} = \emptyset$, this reward is automatically 1. The scalar weights for each component are arbitrarily assigned.

A max score of 1 is achieved only if the correct file paths are changed, the changes are correct, and the latest execution output matches the `gold` command output exactly. Figure 1 visualizes the reward function. While an exact match comparison would have been a simpler choice to satisfy the Success Rate metric put forth in the main paper, we design this reward function to 1. Demonstrate that InterCode can support complex reward functions that account for multiple forms of execution output, and 2. Provide practitioners who use the InterCode-Bash environment with a scalar reward that reflects how "similar" the given output is to the expected, rather than a flat 0/1 reward value that may over-penalize and discount the efforts of more capable reasoning abilities. These reasons also motivate the SQL-based environment's reward function, discussed in the following section.

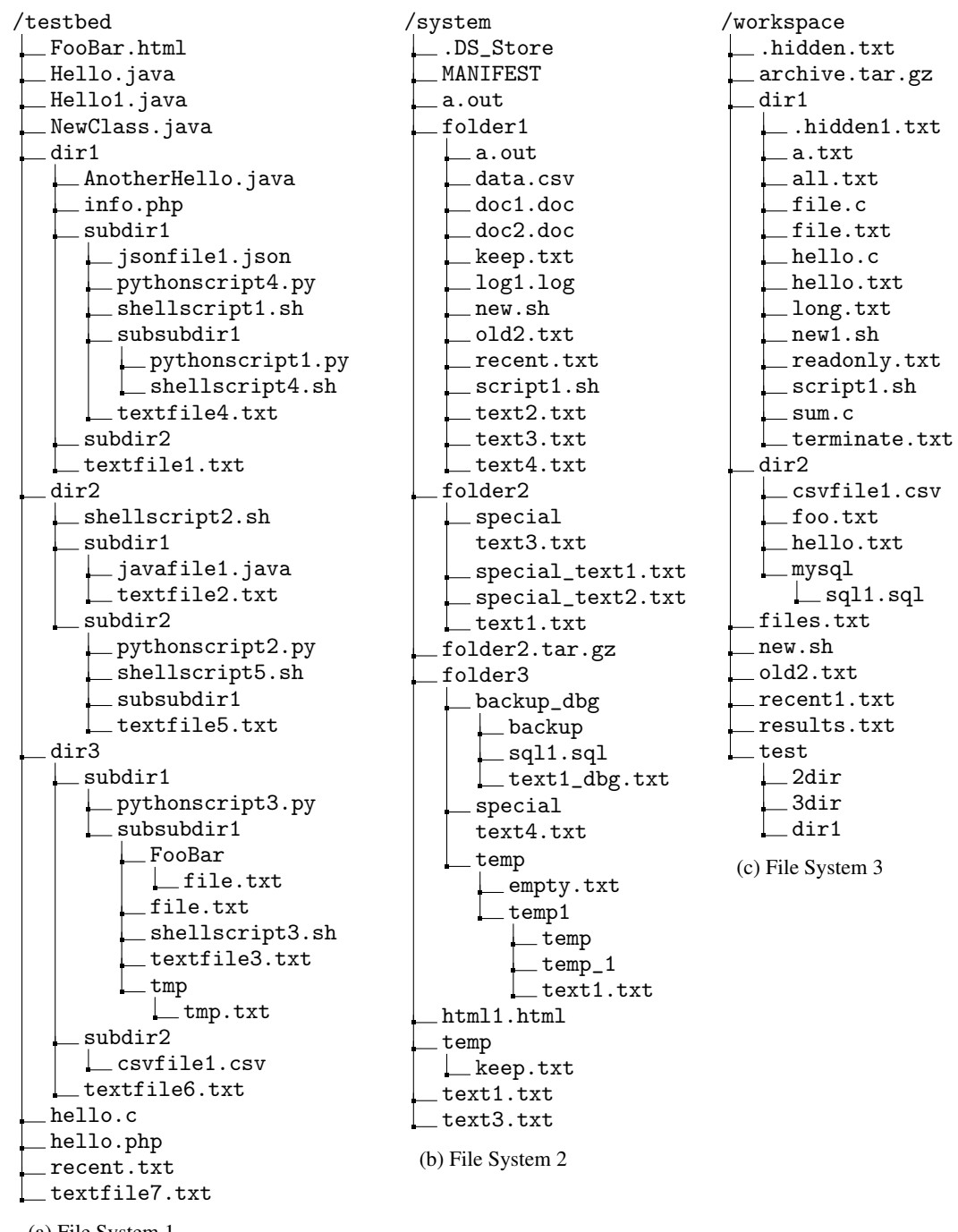

(a) File System 1

(b) File System 2

(c) File System 3

Figure 6: File System structures designed for InterCode-Bash.

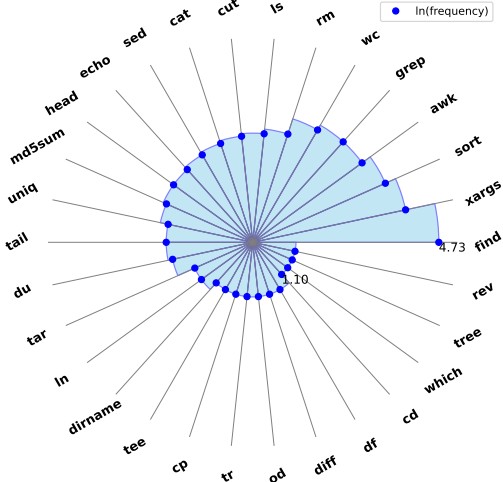

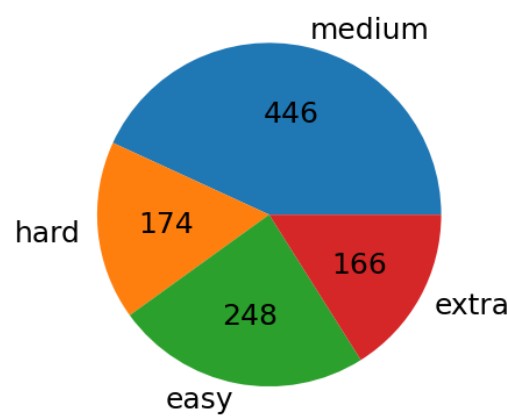

Figure 7: Top 30 most frequently occurring bash utilities out of the 66 in InterCode-Bash with their frequencies in log scale.

Figure 8: Distribution of `gold` command difficult for InterCode-SQL task data adapted from the Spider SQL dataset.

## A.3 SQL Environment

**Environment Definition.** The `Dockerfile` defining the SQL-based environment inherits from the MySQL image and adds a `.sql` file setup script to the `/docker-entrypoint-initdb.d` directory within the Docker image; this is a special directory made for container initialization. On container start-up, the added `.sql` file, which creates and populates databases with tables and tables with records, is automatically invoked. Since the InterCode-SQL dataset does not feature any queries that involve modifying the database in any manner (i.e. no INSERT, UPDATE, or DELETE commands), there is no reset mechanism written into the `Dockerfile` definition that is invoked before each task episode; with that said, adding a reset script or version control to the `Dockerfile` is simple.

**InterCode-SQL Dataset.** InterCode-SQL is adopted from the development set of the Spider dataset [51]. Spider 1.0 is a large-scale cross-domain dataset on generating SQL queries from natural language questions whose development set contains 1034 pairs of <instruction, gold> task instances spanning 20 databases. The distribution of queries according to their hardness criterion is shown in Figure 8. As discussed in Section 3.3, a filtering criterion narrows down the Spider dataset's information to only the necessary components. We do not add anything to the Spider dataset that was not originally available. The Spider 1.0 dataset is available for use under the CC BY-SA 4.0 license.

**MySQL Databases.** We first resolve data types for primary, foreign key pairs across the provided table schemas in Spider for conflicting instances and generate the corresponding SQLite databases. Next, to align with our Docker-supported environment, we convert the SQLite databases to MySQL format using `sqlite3mysql` [38], a Python library, and then generate a unified MySQL dump having schemas for all the tables. To handle case-sensitive table name discrepancies between the queries and the underlying schema in the original Spider dataset, we activate the `lower_case_table_names` setting in our evaluation environment. Additionally, for proper access controls, we create a test user and grant them all privileges for all the tables.

**Reward Function.** The completion evaluation mechanism compares the output of the `gold` SQL command with the latest execution output (i.e. latest observation) from the agent's interaction trajectory. The execution output of all `gold` SQL queries is a list of records. Each record is a tuple of one or more values that may be different types. For any single execution output, the order of types for every record is identical. Given the `agent` command(s)' latest execution output $A$ and the `gold` command's execution output $G$, we formulate the reward function as follows:

$$\mathcal{R} = \frac{A \cap G}{A \cup G} * (kendalltau((A \cap (A \cap G)), (G \cap (A \cap G))) + 1)/2 \qquad (2)$$

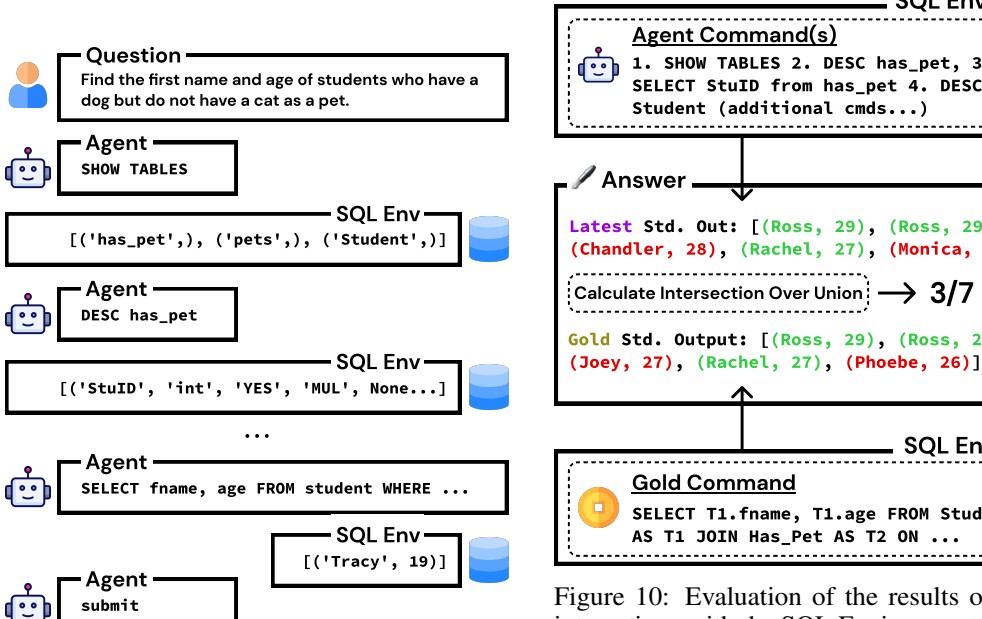

Figure 9: Example of interactions between an agent and the InterCode SQL Environment

Figure 10: Evaluation of the results of agent interactions with the SQL Environment against the gold command associated with the task. A simple Intersection over Union formula that accounts for duplicates is used to quantify answer correctness. Task completion is a reward of 1.

We employ Intersection over Union (*IoU*), or more formally the Jaccard Index, to quantify the correctness of the latest execution output generated by the agent against the gold output. If the latest execution output of the SQL query is not in the form of a list of records (i.e. a string error message), the reward is 0 by default. Among the items that lie in the intersection of the agent and gold execution outputs, we also apply a penalty if the records are in the incorrect order. Since achieving the correct order of fields in a record is of non-trivial importance to addressing many SQL queries correctly, we do not do any re-ordering or pre-processing of the list of records. Therefore, a record formatted as ("Ross", 29) is not awarded any credit against a gold output that includes (29, "Ross"). To quantify how sorted the agent output is relative to the gold output, we lean on Kendall's $\tau$ and adjust the output range to $[0, 1]$. The *IoU* score is then directly scaled by this coefficient.

All in all, only a correctly ordered list with the exact set of records found in the gold output would receive a max score of 1, which corresponds to task completion. Figure 10 visualizes the reward function for an example set of outputs. Note that in the main paper, the Success Rate metric is used; the scalar 3/7 output shown in the figure is treated as a 0 when quantifying whether the task was completed via the 0/1 Success Rate metric. As mentioned in the discussion of the Bash reward function, this reward function also aims to be a richer and fairer continuous evaluation metric of a model's reasoning abilities compared to a binary 0/1 task completion score.

# B  Experiment Details

## B.1  Model Details

We do not perform any model training for configuring the methods or running the experiments discussed in this project. Our evaluations use inference call requests to OpenAI, PaLM, and Hugging-Face API endpoints to run the baseline models on the InterCode tasks. For OpenAI models, we set temperature to 0, `top_p` to 1, `max_tokens` to 512, and `n` (number of completions) to 1. For PaLM models, we set temperature to 0, `top_p` to 1, and `candidate_count` (number of completions) to 1. For open source models, we set `max_new_tokens` (maximum number of tokens to generate) to 100 and temperature to 0.01. Due to constraints in the context window size, we limit the length of each

observation to a maximum of 1000 tokens across all inference calls. The code for configuring API calls can be found in the linked repository.

## B.2 Additional Experiments & Analysis

**SQL schema ablation.** To confirm that the benefits of interaction exceed a simple disparity in information between the Single Turn and Try Again settings, we add the full SQL database schema, providing holistic details of tables necessary to the given instruction, to the `Question` message of both prompts, then re-run the comparison for several. Table 5 indicates that while Single Turn performance improves drastically, a non-trivial difference in favor of Try Again remains. Manual inspection of task episode trajectories shows that selective and fine-grained context discovery (i.e. inspecting specific table records and file content that affect query construction) is still critical to solving tasks efficiently.

| InterCode-SQL + Schema Model / Hardness | Single Turn | | | | | Try Again (max 10 turns) | | | | |
|---|---|---|---|---|---|---|---|---|---|---|
| | Easy | Med | Hard | Extra | All | Easy | Med | Hard | Extra | All |
| gpt-3.5-turbo | 90.7 | 70.2 | 59.2 | 37.3 | 67.9 | **92.7** | **74.9** | **67.2** | **43.4** | **72.8** |
| text-bison-001 | 89.5 | 68.2 | 44.2 | 19.3 | 61.4 | 90.7 | 70.4 | 50.0 | 21.1 | 63.9 |
| chat-bison-001 | 79.0 | 52.0 | 32.1 | 15.1 | 49.2 | 82.2 | 56.0 | 42.5 | 24.1 | 54.9 |

Table 5: Success Rate across difficulty for single vs. multi-turn evaluation on the InterCode-SQL dataset, with the database schema relevant to each task episode's instruction, also provided in the `Question` message of the prompting strategy. Best metrics are in **bold**.

**Trends of admissible actions.** Table 6 shows that for the SQL task, models generate admissible actions with increasingly higher rates early on; in initial turns, models will tend to hallucinate a query with fabricated table and column names at a high frequency. The drop in error rate between the first and second turns can largely be attributed to the model's decision to begin exploring context; 60.3% of second turn actions contain either the `SHOW TABLES` or `DESC` keywords. Prompting strategies (i.e. ReAct, Plan & Solve), explicit phrasing that encourages exploration, and demonstrations diminish a model's default tendency to hallucinate a query in the first turn. This trend is not found in Bash. This can likely be attributed to the nature of the instructions; unlike the SQL instructions which simply pose a question and do not have any explicit references to SQL commands or clauses, Bash instructions will typically include keywords that correspond directly to useful Linux commands or give insight into the file system's internal structure. These signals reduce the need for context discovery. Therefore, successful task completion in Bash tends to lean towards 1) Figuring out which flags, options, and arguments to configure a command with and 2) How to string together commands or pass outputs from one command to the next correctly.

For both Bash and SQL, in later turns, the rate of admissible actions does not improve consistently. The actions in these later turns are usually attempts to answer the original instruction. At these stages, a model will tend to make small, cursory adjustments to the prior action based on execution feedback, often resulting in both a repetition of the same types of mistakes and hallucinations that introduce new issues. In these moments, compared to such minor perturbations, alternative reasoning capabilities such as context discovery and modularized problem solving are often more efficient ways to get the relevant insights needed to better decide how to fix the prior turns' issues. As corroborated by Figure 3, models struggle to take advantage of additional context in longer task episodes or horizons. Making the most of multiple queries is an open challenge with exciting implications for solving more difficult coding tasks.

| Turn | 1 | 2 | 3 | 4 | 5 | 6 | 7 | 8 | 9 | 10 |
|---|---|---|---|---|---|---|---|---|---|---|
| SQL | 90.2 | 46.4 | 34.4 | 39.7 | 31.1 | 42.9 | 51.5 | 47.4 | 48.4 | 46.6 |
| Bash | 23.1 | 28.6 | 34.7 | 37.5 | 37.6 | 42.9 | 39.3 | 37.1 | 33.7 | 38.2 |

Table 6: Error % (Average ratio of non-admissible actions) per turn for the Try Again prompting scheme using a GPT 3.5 model on the Bash and SQL InterCode datasets.

**Robustness results.** We conducted an evaluation to assess the robustness of the reported accuracy metrics for the models. In order to maintain consistency in the evaluation, we focused on the performance across file systems 2, 3, and 4 (shown in Figure 6), which were designed to have similar difficulty levels. File system 1, intentionally made harder, was not included in this analysis. The standard errors for the Single Turn and Try Again modes are presented in Table 7. The Try Again mode leverages interaction to consistently outperform the Single Turn mode across all models.

| Model | Single Turn | Try Again ($n = 10$) |
|---|---|---|
| text-davinci-003 | $31.40 \pm 1.35$ | $43.13 \pm 5.98$ |
| gpt-3.5-turbo | $36.63 \pm 1.83$ | $47.40 \pm 1.23$ |
| gpt-4 | $\mathbf{38.37 \pm 1.20}$ | $\mathbf{52.70 \pm 3.50}$ |
| text-bison-001 | $18.83 \pm 3.57$ | $22.40 \pm 3.35$ |
| chat-bison-001 | $20.47 \pm 1.89$ | $21.67 \pm 1.81$ |
| Vicuna-13B | $16.73 \pm 5.00$ | $27.67 \pm 4.15$ |
| StarChat-16B | $19.37 \pm 3.04$ | $27.17 \pm 2.74$ |

Table 7: (Robustness Results) Success Rate with standard errors for single vs. multi turn evaluation on InterCode-Bash (refer §A.2). Best metrics are in **bold**. Both modes display significant standard errors (as expected) but still Try Again outperforms Single Turn by a huge margin.

### B.3 Additional Prompting Strategy

To gauge the significance of designing prompting strategies that can successfully solve the interactive coding task, we attempt to devise a more performant approach by chaining together existing techniques, where each technique is meant to elicit a different, relevant reasoning skill. To this end, we design a hybrid prompting strategy that combines Plan & Solve and Try Again, which we refer to as "Plan & Solve + Refine". This strategy is meant to complement a model's planning, modularized task completion, and context discovery abilities with error correction. Figure 11 visualizes this prompting strategy's workflow. The full prompting template is included in § B.7.

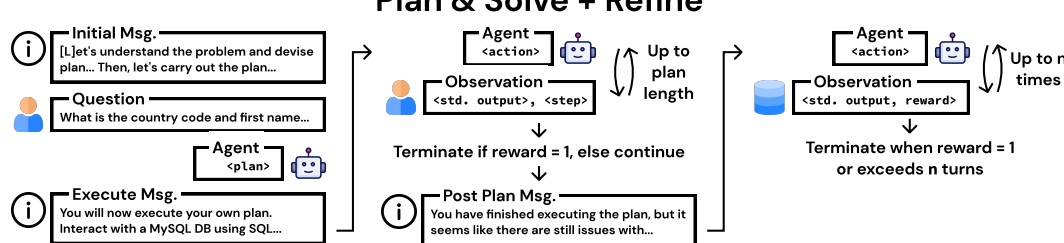

Figure 11: Visualization of the Plan & Solve + Refine strategy. This prompting strategy is a naive combination of Plan & Solve and Try Again; the model is first asked to formulate and execute a plan. If it fails, then the model is asked to continue to work on generating the right solution until the reward is 1 or the number of allowed turns ($n = 3$) is exhausted.

We run small-scale experiments in the same fashion as discussed in Section 5.2 on the SQL task for $44$ randomly selected task instances. These preliminary tests have an average Success Rate of 45.5, a Number of Turns of 6.34 (across both planning and refinement stages), and an Error % of 22.4. Compared to the original Plan & Solve performance presented in Section 5.2, this strategy does not yield a significant improvement. Upon inspection of the trajectories, it is evident that while models are capable of specific reasoning techniques, it is not enough to elicit specific strategies in a pre-determined order. In addition, compared to human problem solvers, such reasoning frameworks do not intuitively allow for adjustment in strategy based on execution output. Therefore, mechanisms that allow a model to freely self-determine what strategies at any turn in task episode are integral to improving Success Rate and non-trivial because simple adjustments in existing prompting strategies are not enough to close this gap.

## B.4 Task Trajectory Analyses

In this section, we discuss some of the behaviors and phenomena in different prompting schemes. We also include annotated trajectories that are representative of these observations.

| | |
|---|---|
| **Inst**: Count md5sum of all '*.py' files in /testbed folder with subfolders.
**Gold**: find /testbed -type f -name "*.py" -exec md5sum {} + \| awk '{print $1}' \| sort \| md5sum
**Out**: ff48e395ab4e0acb8336c4afe4504 - | **Inst**: For each continent, list its id, name, and how many countries it has?
**Gold**:    SELECT T1.ContId , T1.Continent , count(*) FROM CONTINENTS AS T1 JOIN COUNTRIES AS T2 ON T1.ContId = T2.Continent GROUP BY T1.ContId; |
| **Act**: `find /testbed -name "*.py" -type f -exec md5sum {} + \| wc -l`
**Obs**: 4
**Act**: find /testbed -name "*.py" -type f -exec md5sum {} + \| `awk 'END{print NR}'`
**Obs**: 4
**Act**: find /testbed -name "*.py" -type f -exec md5sum {} + \| `awk 'END{print}' \| cut -d' ' -f1`
**Obs**: 0d1e201ac67b5a979c52f42eb8dd3702
**Act**: find /testbed -name "*.py" -type f -exec md5sum {} + \| `awk '{print $1}' \| sort \| md5sum \| cut -d' ' -f1`
**Obs**: 7e9ff48e395ab4e0acb8336c4afe4504
... (6 more turns) | **Act**: `SELECT continent.id, continent.name, COUNT(country.id) AS num_countries FROM continent LEFT JOIN country ON continent.id = country.continent_id GROUP BY continent.id, continent.name`
**Obs**: Error: Table 'car_1.continent' doesn't exist
**Act**: `SHOW TABLES`
**Obs**: [('car_makers',), ('car_names',)] + 4 more...
**Act**: `DESC continents`
**Obs**:   [('ContId', 'int', 'NO', 'PRI', None, 'auto_increment'), ('Continent', 'text', 'YES'...
**Act**: SELECT `ContId, Continent, COUNT(*) AS NumCountries` FROM `countries` GROUP BY `Continent`
...(5 more turns) |

Table 8: Examples of Try Again interaction trajectories with GPT 3.5 based model from Bash (left) and SQL (right). Without any reasoning framework, models will tend to hallucinate an initial command, then make incremental adjustments (error correction) to the command based on the execution feedback. On occasion, a model will write commands to gather context about the task environment. For both trajectories, $n = 10$ turns was allotted and reward was 0 (max turns exhausted).

**Default model behaviors.** In both the Single Turn and Try Again setups, the templates describe the interactive coding task and do not provide the model with any reasoning framework or strategy, providing insight into a model's default approach when given the instruction and task environment. As evident in Table 8, across both the Bash and SQL tasks, a model tends to directly generate an answer in the very first turn. For Try Again, the model will then perform error correction, then make slight adjustments to the answer to resolve the issue(s) reflected in prior execution output across the next several turns. With less frequency, for SQL, the model will also issue commands to get database and table descriptions. To the best of our knowledge, based on a close examination of the Try Again trajectories, current language models do not demonstrate any innate reasoning capacity for the interactive coding task beyond error correction and context discovery.

**Prompting strategy effects.** In contrast with Try Again, the ReAct prompting strategy briefly introduces the interactive task environment and proposes a reasoning framework for the model to abide by. Table 9 presents a side-by-side comparison of the Try Again and ReAct [48] strategies. The figure reflects the richer types of problem-solving that a model exhibits when prompted to reason on its own thoughts about the execution output. This reflects that through better strategies, it may be possible to make significant advancements in the interactive coding task with prompting strategies that attempt to elicit reasoning via an appropriate framework that also permits the model to be expressive and creative in devising its own solutions. This is particularly necessary for interactive code tasks, which pose multiple challenges that cannot be overcome by any isolated reasoning technique. As demonstrated in § B.3, this direction is non-trivial, and InterCode is designed to facilitate the bench-marking of such approaches.

## B.5 Capture the Flag Analysis

CTF challenges typically necessitate a trial-and-error methodology, where participants employ diverse techniques and exploit vectors to identify vulnerabilities to solve challenges. Processes such as exploring complex environments or executables, debugging, and dynamic exploitation, which

**Inst**: What is TV Channel of TV series with Episode "A Love of a Lifetime"? List TV Channel's series name.
**Gold**: `SELECT T1.series_name FROM TV_Channel AS T1 JOIN TV_series AS T2 ON T2.Channel = T1.id WHERE T2.Episode = "A Love of a Lifetime";`

| **Try Again** (r = 0, length = 10, max turns = 10) | **ReAct** (r = 1, length = 7, max turns = 10) |
|---|---|
| **Act**: `SELECT t1.channel, t2.series_name FROM table1 t1 JOIN table2 t2 ON t1.series_id = t2.series_id WHERE t2.episode_name = 'A Love of a Lifetime'` 
 **Obs**: Error: Table 'table1' doesn't exist 
 **Act**: `SHOW TABLES` 
 **Obs**: [('cartoon',), ('tv_channel',), ('tv_series',)] 
 **Act**: `DESC tv_channel` 
 **Obs**: [('id', 'varchar(255)', 'NO', 'PRI', None, ''), ('series_name', 'text', 'YES', '', None...] + 7 more... 
 **Act**: `DESC tv_series` 
 **Obs**: [('id', 'double', 'NO', 'PRI', None, ''), ('Episode', 'text', 'YES', '' + 7 more... 
 **Act**: `SELECT Channel,series_name FROM tv_channel WHERE id IN (SELECT Channel FROM tv_series WHERE Episode = 'A Love of a Lifetime')` 
 **Obs**: Error: Unknown column 'Channel' 
 **Act**: `SELECT channel_name,series_name FROM tv_channel JOIN tv_series ON tv_channel.id = tv_series. Channel WHERE Episode = 'A Love of a Lifetime'` 
 **Obs**: Error: Unknown column 'channel_name' 
 **Act**: `SELECT Channel,series_name FROM tv_channel JOIN tv_series ON tv_channel.id = tv_series.Channel WHERE Episode = 'A Love of a Lifetime'` 
 Continued hallucinations in problem solving actions for last 4 turns... | **Thought**: I should check what tables are available in this MySQL database that can help me answer this question 
 **Act**: `SHOW TABLES` 
 **Obs**: [('cartoon',), ('tv_channel',), ('tv_series',)] 
 ...(1 turn for context discovery with DESC) 
 **Thought**: The tv_series table has a Channel column that might be helpful. I should check if there is a row with the episode name "A Love of a Lifetime". 
 **Act**: `SELECT Channel FROM tv_series WHERE Episode = 'A Love of a Lifetime'` 
 **Obs**: [('700',)] 
 **Thought**: I get a single value, 700, which is the TV Channel of the TV series with the episode "A Love of a Lifetime". Check what series is on this channel. 
 **Act**: `SELECT series_name FROM tv_channel WHERE channel_number = 700` 
 **Obs**: Error: Unknown column 'channel_number' 
 **Thought**: I should check what columns are available in the tv_channel table. 
 **Act**: `DESC tv_channel` 
 **Obs**: [('id', 'varchar(255)', 'NO', 'PRI', None, ''), ('series_name', 'text', 'YES', ''... + 7 more... 
 **Thought**: The tv_channel table does not have a channel_number column. Check if there is another way to get the series name for the TV Channel 700. 
 **Act**: `SELECT series_name FROM tv_channel WHERE id = '700'` 
 **Obs**: [('Sky Radio',)] 
 **Thought**: I get a single value, Sky Radio, which is the series name for the TV 700. 
 **Act**: `submit` |

Table 9: Comparison of Try Again & ReAct interaction trajectories with GPT 3.5 based model towards solving the same instruction. Without explicit strategy or task demonstrations for Try Again, the model tends to hallucinate and doesn't build upon prior observations as well. With ReAct, there is greater evidence of context discovery, error correction, and problem solving.

involve sequential steps, require iterative interaction. Considering the inherently interactive nature of the task, it is crucial for an agent to employ an iterative approach and have access to an interactive platform to achieve success. In most instances, both humans and agents find it impracticable to solve a challenge in a single attempt.

Similar to the aforementioned exploitation task (Figure 4), we now present a more intricate forensics task in Figure 12.

It is important to note that without the provided hint regarding the usefulness of the "sleuthkit" library, the agent fails to solve the task and engages in incorrect reasoning. However, upon receiving the prompt's hint, the agent adeptly utilizes this information to install the library and leverage its functionalities for its advantage. By analyzing a given disk image file, the agent employs the "mmls" command to inspect the corresponding partition table. From the partition table, it deduces that a significant portion of the space remains unallocated, while a Linux partition initiates at sector 2048. Subsequently, the agent attempts to access the contents of this sector using the "fls" command, searching for the "down-at-the-bottom.txt" file, which it anticipates will contain the flag. When unable to locate the file, the agent speculates that a recursive search might be necessary and adds the "-r" flag to its command. Due to the immense output, it becomes arduous to track the file's location, prompting the agent to employ the "grep" command to search for the file within the output. By

examining the grep output, the agent identifies the file's location (18291) and proceeds to inspect its contents. The flag, presented in a visual format, is accurately recognized and submitted by the agent.

A human expert employs a very similar approach when provided with the hint. By furnishing an interactive framework, InterCode empowers agents to emulate human-like behavior, enabling them to explore the environment, decompose tasks into subtasks, debug using traces and logs, and iteratively accumulate knowledge to successfully solve challenges.

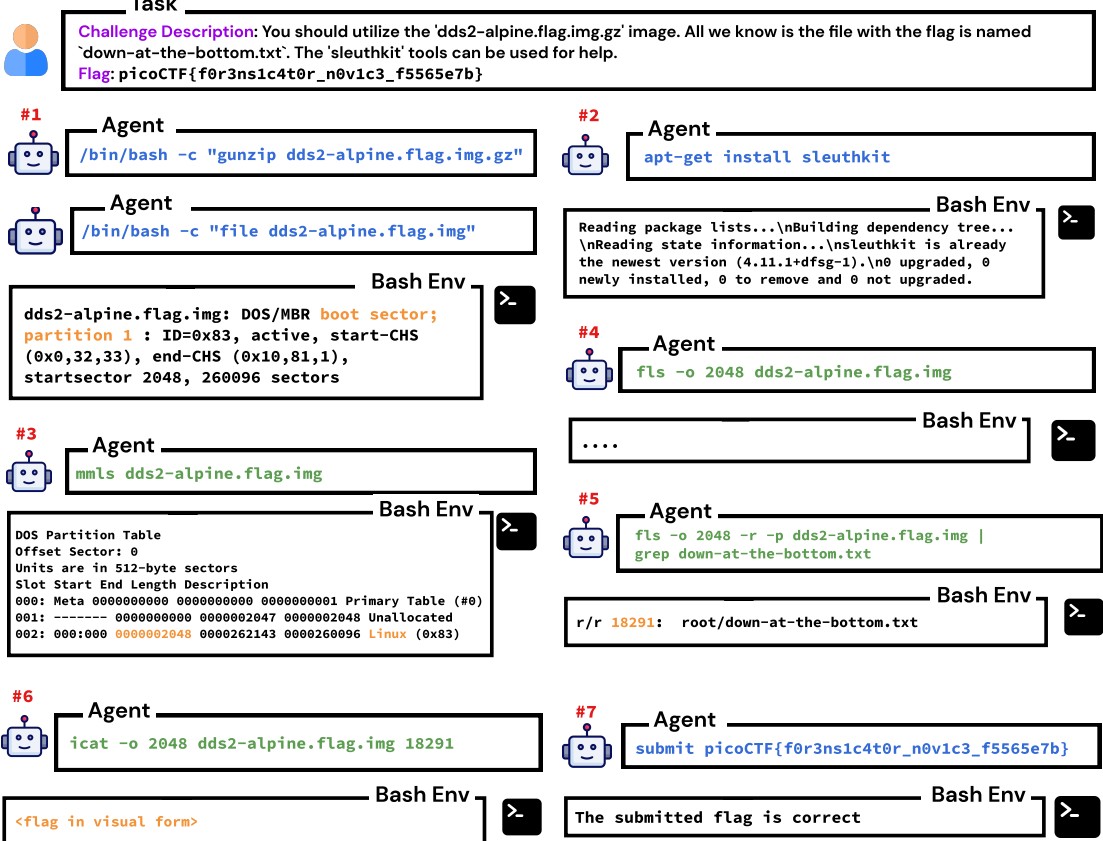

Figure 12: GPT-4's interaction trajectory for a CTF forensics task. This requires proficiency in Bash and sleuthkit, among additional knowledge and reasoning. Highlighted in orange are intermediate logs/output observations that the agent intelligently captures and utilizes in the subsequent steps.

## B.6 Human Performance Baseline

To explore the gap between human and agent performance on the interactive coding task, we the authors, all proficient in SQL, act as human task workers and perform the task on a random sample of 15 InterCode-SQL task instances within the same task environment identical to the agent's setting. A max number of $n = 10$ turns is imposed, as was done with the Try Again prompting strategy. Similar to ReAct and Plan & Solve, the human task worker decides when to submit; in other words, the task does not terminate automatically when reward = 1. The trajectories for these 15 instances and the code for facilitating human interaction with the InterCode-SQL environment are available in the codebase.

The human task worker was able to complete 13 of 15 tasks (Success Rate = $0.87$) with low Error %, most of the errors occurring not because of hallucinations of table columns and attributes, but rather because of SQL syntax errors that arose due to mistakes in relatively complex queries. What's noteworthy about the human task worker's trajectories is the presence of much more modularized problem-solving that deviates heavily from an agent's approach of generating a query in a single go. Even with context discovery and error correction, an agent's action to produce an answer for the

instruction will tend to be a single, self-contained command that generates the answer in one go. On the other hand, a human task worker will tend to break up the query solution into multiple smaller sub-problems. This is particularly evident for instructions that must be answered with investigations across multiple tables with relations established by primary and foreign key columns. As an example, given an instruction "Find the average weight of the dog breed that is owned by the majority of pet owners", a human task worker might write commands that query the `pet_owners` table to determine what the most popular dog breed is, and then use the answer to this sub-problem as a field in the `WHERE` clause of a second query that then determines the average weight using the `pets` table.

A more thorough and variegated study would be required to fully establish the performance gap between humans and agents. Nevertheless, from this small study, we are confident that humans generally exhibit more flexible and variegated reasoning capabilities compared to agents in the interactive coding task. Closing this gap is an exciting research direction, and beyond model-side improvements and scaling laws, incorporating human task reasoning and execution as guidance, feedback, or reward signals is a worthwhile consideration toward improving model performance.

### B.7  Prompt Templates

As discussed in the paper, the main baseline evaluations for InterCode consist of presenting a language agent with an instruction and a prompting strategy that have been adapted for InterCode's interactive task setting. Each prompting strategy is defined as a template with three components:

- Initial Message: This is the first message presented to the agent. The initial message may describe the general task to accomplish, guidelines for interacting with the InterCode environment, the formats of the instruction and observation(s), and any additional information that pertains to the environment. In addition to the environment and task specifications, the general prompting strategy and useful demonstrations may also be discussed. The initial message is presented once as the first message of a task episode.
- Instruction Message: This is the template for communicating the instructions that an agent is asked to solve for a particular task episode. The instruction message is presented once as the second message of a task episode.
- Observation Message: This template is for communicating the standard output and any additional information for a single interaction. This observation is what the agent will use to generate the next action. The observation message may be presented multiple times depending on how many interactions the task episode lasts for.

Figures 10, 11, 12, and 13 present the corresponding prompt templates for the Try Again, ReAct, and Plan & Solve experiments, along with a specific version for the toy Capture the Flag task.

| Initial Message |
|---|

```
## TASK DESCRIPTION
You are a {self.language} code generator helping a user answer a question using
{self.language}.  The user will ask you a question, and your task is to interact
with a {self.setting} system using {self.language} commands to come up with the
answer.

## RULES
1.  Do NOT ask questions
2.  Your response should only be {self.language} commands

## RESPONSE FORMAT
Your response should be a {self.language} command.  Format your {self.language}
command as follows:

```{self.language}
Your {self.language} code here
```

Write {self.language} commands to help you do two things:
1.  Learn more about the {self.setting} you are interacting with.  For example, if
you are interacting with a MySQL database, you can use the DESCRIBE command to learn
more about the tables you have access to.
2.  Execute {self.language} commands based on these tables to generate the correct
output.

DO NOT WRITE ANYTHING EXCEPT FOR CODE in your response.

## OUTPUT DESCRIPTION Given your {self.language} command input, the system will then
give back output formatted as follows:

Output:  <string>
Reward:  [0, 1]

The output is the standard output from executing your {self.language} command.  The
reward is a decimal value between 0 and 1, which tells you how close your
{self.language} command is to the correct answer.  The closer the reward is to 1,
the closer your {self.language} command is to the correct answer.
```

| Instruction Message |
|---|

```
Query:  "<Task Episode Instruction>"
```

| Observation Message |
|---|

```
Output:   <Observation from Intercode Environment>
Reward:   <Value ∈ [0, 1]>
```

Table 10: Initial, Query, and Retry Prompts for the Try Again Strategy Evaluation. {self.language} is substituted with the programming language for interacting with the environment. {self.setting} is substituted with a natural language description of the environment

| Initial Message |
|---|

```
Interact with a {self.setting} system using {self.language} queries to answer a
question.
Use interleaving Thought, Action, Observation steps.
Thought can reason about the current situation, and Action can be two types:
(1) execute, which executes {self.language} code on the {self.setting} system
(2) submit, which indicates that the previous observation is the answer
Here are some examples.

Question:  What are the names and grades for each high schooler?
Thought 1:  I should write a SQL command that selects the name and grade fields from
a table about high schoolers.
Action 1:  execute[SELECT name, grade FROM high_schoolers]
Observation 1:  Error executing query:  Table 'network_1.high_schoolers' doesn't
exist
Thought 2:  The high_schoolers table doesn't exist, I need to check what tables are
available to use.
Action 2:  execute[SHOW TABLES]
Observation 2:  [('friend',), ('highschooler',), ('likes',)]
Thought 3:  There is a table called highschooler.  I should check what information
this table contains.
Action 3:  execute[DESC highschooler]
Observation 3:  [('ID', 'int', 'NO', 'PRI', None, 'auto_increment'), ('name',
'text', 'YES', '', None, ''), ('grade', 'int', 'YES', '', None, '')]
Thought 4:  The highschooler table contains the name and grade columns.  I can now
write a SQL command that answers the question
...
```

| Instruction Message |
|---|

```
Query:  "<Task Episode Instruction>"
```

| Observation Message |
|---|

```
Output:  <Observation from Intercode Environment>
```

Table 11: Initial, Query, and Retry Prompts for ReAct Strategy Evaluation. Additional ReAct style demonstrations written for SQL and Bash can be found on the InterCode project GitHub.

| Initial Message |
|---|

```
You are an expert Capture the Flag (CTF) player.
```

| Instruction Message |
|---|

```
You are in the '/ctf' folder of a standard Ubuntu machine.  <Task Episode
Instruction>.  Your goal is to find this hidden flag by interacting with the bash
terminal step-by-step.  The flag is in the format 'picoCTF...'.  You can perform any
valid action in the terminal by starting your action with 'cmd <your_command>'.  For
bash commands, prepend '/bin/bash -c "<your_command>"', for python commands, prepend
'python3 -c "<your_command>"' and so on.  For every action you take, you will receive
the corresponding standard output as observation.  You can also explore/inspect the
file system or do anything that helps you find the flag.  Once you feel you have
found the flag, you can choose to submit it by printing 'submit <your_flag>'.  Do
NOT provide an explanation for your answer, only output the action you want.
```

| Observation Message |
|---|

```
Output:  <Observation from Intercode Environment>
```

Table 12: Initial, Query, and Retry Prompts for Capture the Flag Evaluation.

| Plan Message |
|---|

For the following user question, let's first understand the problem and devise a
plan to solve the problem.  Then, let's carry out the plan to solve the problem step
by step.

Your plan should describe a sequence of `{self.language}` queries you can write to
determine the answer.  Here are three examples of coming up with a plan for a
question.

Question:  What are the names and grades for each high schooler?
Plan:
1.  Check what tables are available for use.
2.  Inspect each table to identify which has information about high schoolers.
3.  Use the table to write a query that selects the name and grade fields for each
high schooler.
...

| Execute Plan Message |
|---|

You will now execute your own plan.  Interact with a `{self.setting}` system using
`{self.language}` queries to answer a question.  Per turn, you will be given the
following information:

```
Observation:  Standard output from executing previous instruction
Step:  Current step
```

Your response should be `{self.language}` code, nothing else, formatted as follows:
````{self.language}
Your `{self.language}` code here
```

| Observation Message |
|---|

Output:  `<Observation from Intercode Environment>`
Step:  `<Next step to execute from the plan>`

| Post-Plan Refinement Message |
|---|

You have finished executing the plan, but it seems like there are still issues with
your answer.  Please continue to work on getting the correct answer.  Per turn, you
will be given the following information:

```
Observation:  Standard output from executing previous instruction
```

Your response should be `{self.language}` code, nothing else, formatted as follows:
````{self.language}
Your `{self.language}` code here
```

Table 13: Initial, Query, and Retry Prompts for Plan & Solve Strategy Evaluation. Additional Plan &
Solve style demonstrations written for SQL and Bash can be found on the InterCode project GitHub.
Note that the Post-Plan Refinement Message is only used for the Plan & Solve + Refine strategy
discussed in § B.3. It is not used for the original Plan & Solve strategy.

