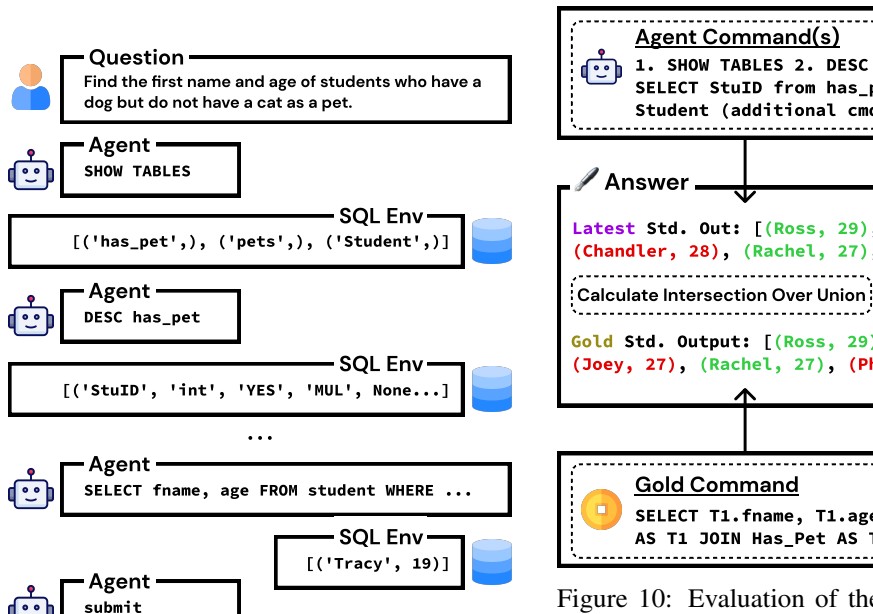

**Question** Find the first name and age of students who have a dog but do not have a cat as a pet.

**Agent** `SHOW TABLES`

**SQL Env** `[('has_pet',), ('pets',), ('Student',)]`

**Agent** `DESC has_pet`

**SQL Env** `[('StuID', 'int', 'YES', 'MUL', None...]`

...

**Agent** `SELECT fname, age FROM student WHERE ...`

**SQL Env** `[('Tracy', 19)]`

**Agent** `submit`

Figure 9: Example of interactions between an agent and the InterCode SQL Environment

**SQL Env — Agent Command(s)**
`1. SHOW TABLES 2. DESC has_pet, 3. SELECT StuID from has_pet 4. DESC Student (additional cmds...)`

**Answer**
`Latest Std. Out: [(Ross, 29), (Ross, 29), (Chandler, 28), (Rachel, 27), (Monica, 25)]`

`Calculate Intersection Over Union` → 3/7

`Gold Std. Output: [(Ross, 29), (Ross, 29), (Joey, 27), (Rachel, 27), (Phoebe, 26)]`

**SQL Env — Gold Command**
`SELECT T1.fname, T1.age FROM Student AS T1 JOIN Has_Pet AS T2 ON ...`

Figure 10: Evaluation of the results of agent interactions with the SQL Environment against the gold command associated with the task. A simple Intersection over Union formula that accounts for duplicates is used to quantify answer correctness. Task completion is a reward of 1.

**Reward Function.** The completion evaluation mechanism compares the output of the `gold` SQL command with the latest execution output (i.e. latest observation) from the agent's interaction trajectory. The execution output of all `gold` SQL queries is a list of records. Each record is a tuple of one or more values that may be different types. For any single execution output, the order of types for every record is identical. Given the `agent` command(s)' latest execution output $A$ and the `gold` command's execution output $G$, we formulate the reward function as follows:

$$\mathcal{R} = \frac{A \cap G}{A \cup G} * (kendalltau((A \cap (A \cap G)), (G \cap (A \cap G))) + 1)/2 \qquad (2)$$

We employ Intersection over Union (*IoU*), or more formally the Jaccard Index, to quantify the correctness of the latest execution output generated by the agent against the `gold` output. If the latest execution output of the SQL query is not in the form of a list of records (i.e. a string error message), the reward is 0 by default. Among the items that lie in the intersection of the `agent` and `gold` execution outputs, we also apply a penalty if the records are in the incorrect order. Since achieving the correct order of fields in a record is of non-trivial importance to addressing many SQL queries correctly, we do not do any re-ordering or pre-processing of the list of records. Therefore, a record formatted as (`"Ross"`, `29`) is not awarded any credit against a `gold` output that includes (`29`, `"Ross"`). To quantify how sorted the `agent` output is relative to the `gold` output, we lean on Kendall's $\tau$ and adjust the output range to $[0, 1]$. The *IoU* score is then directly scaled by this coefficient.

All in all, only a correctly ordered list with the exact set of records found in the `gold` output would receive a max score of 1, which corresponds to task completion. Figure 10 visualizes the reward function for an example set of outputs. Note that in the main paper, the Success Rate metric is used; the scalar $3/7$ output shown in the figure is treated as a 0 when quantifying whether the task was completed via the 0/1 Success Rate metric. As mentioned in the discussion of the Bash reward function, this reward function also aims to be a richer and fairer continuous evaluation metric of a model's reasoning abilities compared to a binary 0/1 task completion score.

## B  Experiment Details

### B.1  Model Details

We do not perform any model training for configuring the methods or running the experiments discussed in this project. Our evaluations use inference call requests to OpenAI, PaLM, and Hugging-Face API endpoints to run the baseline models on the InterCode tasks. For OpenAI models, we set temperature to 0, `top_p` to 1, `max_tokens` to 512, and n (number of completions) to 1. For PaLM models, we set temperature to 0, `top_p` to 1, and `candidate_count` (number of completions) to 1. For open source models, we set `max_new_tokens` (maximum number of tokens to generate) to 100 and temperature to 0.01. Due to constraints in the context window size, we limit the length of each observation to a maximum of 1000 tokens across all inference calls. The code for configuring API calls can be found in the linked repository.

### B.2  Additional Experiments & Analysis

**InterCode-SQL additional results.** Table 5 includes results for additional experiments on open-source models that were completed by the supplementary deadline. Again, the advantage of interactions offered in the Try Again scenario accounts for a multi-point disparity in successful task completion for the SQL task across all degrees of difficulty.

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

## C  Data Collection Risks

The transformations performed to the NL2Bash [29] and Spider [51] datasets generally involve removing instructions with gold commands that are not supported by the given task environment, grounding instructions and commands to the environment, and removing unnecessary fields provided by the original dataset from the version adapted to InterCode. Given this technically based re-purposing of the dataset, we believe that these changes do not introduce any new risks that were not present in the original dataset.

The human trajectories discussed in § B.6 are a small-scale study that again, was performed by the authors to gauge the performance gap between large language models and experts. These trajectories are available in the linked repository and created from the same logging mechanism that was used for the experiments performed on base models with different prompting strategies. The trajectories do not capture any personal information. With that said, given that these trajectories are the product of a small set of individuals, the problem-solving strategy reflected across trajectories may be biased towards some techniques and lean less heavily on others. Approaches that attempt to leverage human feedback and guidance toward training or tuning code models and language models should be founded on more extensive and thorough human demonstration data collection.

## D  Potential Societal Impacts & Limitations

InterCode's goal of formulating tasks to advance the development decision-making and code agents is an exciting research direction that also warrants concerns about safety and fairness.

**Coverage of languages.** The InterCode codebase currently features two tasks with Bash and SQL programming languages as action spaces. We plan to expand the number of InterCode based tasks to cover more programming languages as further demonstrations of the `InterCodeEnv`'s utility along with improving InterCode's ease of use for practitioners interested in InterCode as a training platform. As well as programming languages, additional ongoing work also aims to feature more datasets, task environments, and types of tasks.

**Limitations of the CTF task.** The Capture the Flag toy dataset showcases InterCode's serviceability for developing novel tasks with new code understanding challenges that can easily be used for training and evaluating models. With that said, this demonstration currently only has four task instances. We hope to put forth a more thorough examination of the Capture the Flag task's challenges, provide a clearer picture of the performance of existing models on this task, and release a more comprehensive dataset.

**Training agents with InterCode.** While InterCode in its current state can be used as a training platform for decision-making code agents, the existing codebase does not currently include any examples of training code that uses InterCode in this manner. This is a direction we are interested in pursuing shortly. The InterCode task formulation and usage of the Gym API naturally lends itself to

use for creating decision-making agents that can leverage techniques such as reinforcement learning or imitation learning.

**Safety of developing code agents.** InterCode's use of Docker containers ensures the safe execution of commands in a realistic simulated task environment. With this said, the Bash and SQL InterCode environments currently do not explicitly impose any strict limitations on the action space. While the execution of irreversibly detrimental commands is mitigated by Docker, a direct sim-to-real transfer of an InterCode-trained agent to a real system may put the system at risk. To combat this, the `InterCodeEnv` interface allows task designers to add their own execution logic that can provide guardrails on model behavior and define an allow-list of permissible commands to eliminate the risk of potentially disastrous commands.