# OpenReview forum: "InterCode: Standardizing and Benchmarking Interactive Coding with Execution Feedback"
_NeurIPS.cc/2023/Track/Datasets_and_Benchmarks — NeurIPS 2023 Datasets and Benchmarks Poster_

### Official Review · Reviewer_yCTe · 2023-07-19
**A novel benchmark for studying advanced capabilities of LLM**

**Rating:** 8
**Confidence:** 3
**Correctness:** The dataset is constructed and evalua…
**Clarity:** The paper is written clearly and easy…

**Strengths:**

* The interactive evaluation benchmark for code is novel and interesting. The InterCode framework is well implemented to be easy to use, extensible, and safe.
* Evaluation on InterCode demonstrates LLMs' capability in error correction and context discovery, which hasn't been well quantified previously.
* The paper and the appendix provide plenty of interesting examples of how LLMs solve a problem by interacting with the environment.

**Additional Feedback:**

* In figure1, how does the agent know the final output file should be named "concated.txt"? If the agent used a different filename, would it receive a score of 0?
* I think Appendix B.2 Table 6 is an important baseline because previous text-to-sql approaches often include table schemas as part of the input. The results also show interesting patterns, such as harder problems benefiting more from multi-turn prompting. It's worth mentioning with a pointer in the main paper.

**Documentation:**

Dataset collection and usage are well documented with examples.

**Ethics:**

I do not see ethical concerns.

**Limitations:**

The authors have adequately addressed all the limitations.

**Opportunities For Improvement:**

* Covering more languages will strengthen this work. It would be ideal to include at least one general-purpose programming language such as Python/Java. The paper mentions this is part of the ongoing work.

**Relation To Prior Work:**

Relation to prior work is clearly discussed. While there have been some previous efforts on interactive code evaluation, InterCode is the first general framework that is not constrained by task, language, or action space.

**Summary And Contributions:**

This paper presents InterCode, an interactive coding benchmark with execution feedback. An agent can interact with the InterCode environment by executing the generated response, receiving feedback, and refining the answer iteratively, which enables more diverse problem-solving paradigms than traditional static tests. The authors have implemented two test sets in Bash and SQL, while the platform is language-agnostic and extensible. Execution happens inside a docker to ensure safety.

The paper further evaluates several instruction-tuned LLMs on InterCode. Results show that with multi-turn prompting and interactive feedback, models exhibit the capability of error correction and context discovery, which significantly improves their performance compared to the single-turn baseline.

---

> ### Author Response · Authors · 2023-08-18
>
> Thanks for finding InterCode novel and interesting, and providing careful comments!
>
> ### 1. Increasing Coverage of Supported Languages
> Please see **General Response 1 and 2**, where we expand InterCode to Python and CTF with new experiments, and discuss how to support C++/Java.
>
> ### 2. Figure 1 Critique
> We should have specified the name of the final output file in the query, thanks for the catch! However, by construction, our dataset has the absolute paths to files/folders and names of resultant files/folders to be used/created specified in the query itself to avoid such ambiguities. This case would not arise in practice.
>
> The corresponding query in the dataset is:
> - "query": "Concatenate all .txt files residing in the /system tree into a single file \"/system/folder3/temp/empty.txt\"",
> - "gold": `find /system -name \"*.txt\" -exec cat {} \\; > /system/folder3/temp/empty.txt"`
>
> Nevertheless, as long as the bash logic is correct and the agent used a different file name (than what is specified in the query) is incorrect we would have a non-zero reward as our reward function checks for similarity and correctness as well along with file-system correctness.
>
> ### 3. SQL Table Schema Experiment Reference
> We’ll revise the paper and add a line in section 5.1 that points to Appendix B.2 Table 6, stating that interaction is beneficial even when full table descriptions are offered. Thanks for pointing this out.

---

> > ### Comment · Reviewer_yCTe · 2023-08-24
> >
> > Thanks for the response and the clarification about Fig1. I'll keep my score.

---

### Official Review · Reviewer_14Ca · 2023-07-21
**Promising direction for code generation research**

**Rating:** 7
**Confidence:** 3
**Correctness:** Yes.
**Clarity:** Yes.

**Strengths:**

Mentioned in Summary And Contributions.

**Additional Feedback:**

N/A

**Documentation:**

Yes.

**Opportunities For Improvement:**

The framework currently only supports Bash and SQL, which are easy to get instant feedback. How can higher-level languages like Python, Java and C++ be supported?
Besides code generation, are there any other tasks that can potentially benefit from this framework?

**Relation To Prior Work:**

Yes.

**Summary And Contributions:**

This paper introduces a framework that supports interactive execution-based feedback for coding agents. Coding in reality is interactive in nature. Developers frequently execute the code while developing and closely monitors the diagnostic feedbacks during runtime to update the code accordingly. However, existing code generation methods usually do not support the interactive manner, which introduces misalignment in the task that potentially affects performance. The framework proposed in this paper, InterCode, facilitates a promising direction for code generate research, in which an agent interacts with runtime environment and get diagnostic feedbacks in real-time. This setting is specifically beneficial for reinforcement learning of code generation models.

---

> ### Author Response · Authors · 2023-08-18
>
> ### 1. Supporting higher level languages
> Please see **General Response 1 and 2**, where we expand InterCode to Python and CTF with new experiments, and discuss how to support C++/Java.
>
>
> ### 2. Other tasks that may benefit from InterCode
> This is an interesting open question. One observation is that **a lot of other interactive tasks can be readily converted to (Python) code interaction**, e.g. Python API interactions with search engine for question answering or shopping websites for web navigation [1],  Python code as interactive control polices for robots [2], etc.
>
> [1] ReAct: Synergizing Reasoning and Acting in Language Models.
> [2] Code as Policies: Language Model Programs for Embodied Control.

---

> ### Author Response · Authors · 2023-08-25
> **Look forward to discussion**
>
> Thanks again for your time and feedback!
>
> As the rebuttal is ending soon, we'd like to see if your concerns have been addressed, and would be happy to engaging in further discussions if needed!

---

> > ### Comment · Reviewer_14Ca · 2023-08-29
> >
> > Thank the authors for the discussion. I will maintain my ratings.

---

### Official Review · Reviewer_AK1E · 2023-07-22
**Review for InterCode, a framework for creating interactive code enviornments**

**Rating:** 7
**Confidence:** 4
**Correctness:** Yes. But I didn't check thoroughly.
**Clarity:** Yes

**Strengths:**

1.Interactive code generation and code grounding are important research directions for the real-world application of code generation and this benchmark makes further progress towards standardizing and benchmarking these directions.

2.This benchmark demonstrates its potential in future code generation tasks, such as flexibly using multiple languages to accomplish more complex tasks. This makes the agent behave more like a human rather than just a code completion tool, which is really exciting.

**Additional Feedback:**

Questions
1.Can Intercode contain tasks like “reverse engineering” in [1] Section C.4? Some example demonstrations would be good.

2.In Figure 3, why does the number of interaction turns have a much more significant impact on SQL than on Bash?

[1] Sparks of Artificial General Intelligence: Early experiments with GPT-4


**Documentation:**

Yes. But I didn't check thoroughly.

**Opportunities For Improvement:**

1.Some related work aspect has been neglected. For example, WOB [1] and MiniWoB++ [2-3] also proposed benchmarks to transform natural language queries into a series of instructions through interaction and modeled the entire problem as a classical reinforcement learning environment. I think the main difference may lie in the application scenarios, where WOB focuses mainly on web interactions, while Intercode provides a platform for more general application scenarios. However, this kind of comparison is necessary.

2.The benchmark may not be comprehensive enough. For example, there are only two main languages, and the number of problems for each language is relatively small, lacking a process for generating questions automatically. Moreover, in the bash task, the system directories seem to be predefined, lacking diversity. Also, I'm curious how intercode measures compiled languages like C++ since the paper claims that it can “can have any programming language(s) as the action space”.

3.Some experiment settings are unclear. For example, why weren’t Vicuna and Starchat tested on SQL bench?

[1] World of Bits: An Open-Domain Platform for Web-Based Agents
[2] Reinforcement Learning on Web Interfaces using Workflow-Guided Exploration
[3] Language Models can Solve Computer Tasks

**Relation To Prior Work:**

More discussion would be better.

**Summary And Contributions:**

The paper presents InterCode, a framework for creating interactive code environments with execution feedback. InterCode allows an agent to generate code in response to natural language instructions and receive feedback from the execution environment. The framework is language and platform agnostic, uses Docker containers for safe execution, and supports different types of reward functions. The paper evaluates several state-of-the-art language models on two interactive code tasks with Bash and SQL as action spaces and shows the benefits of interaction and different prompting strategies. The paper also demonstrates how InterCode can be used to create new and challenging tasks such as Capture the Flag, a coding puzzle that requires multiple skills and reasoning abilities. InterCode is released as a new benchmark for advancing code understanding and generation capabilities.

---

> ### Author Response · Authors · 2023-08-18
>
> ### 1. Related Work
> Thanks for the good suggestion! We will discuss these related work on interactive language environments.
>
> ### 2. Improving Benchmark Comprehensiveness
> - Only two main language: see **General Response 1**, where we expanded InterCode to Python and CTF with new experiments.
> - Generating questions automatically: this is an important open problem in the field, and probably orthogonal to our contributions. However, as most real-world coding activities (e.g. Github commits) are interactive, incremental, and driven by execution feedback (e.g. unit tests or reported errors), the InterCode setup might be at least more useful for *collecting* new problems and solutions in a more scalable way (a followup direction we are looking into).
> - Bash environment diversity: Appendix A.2 contains details on how we construct file systems for NL2Bash. A significant portion of the bash tasks’ NL instructions refer to paths (i.e. “Move directories in path “testbed/dir1/subdir1” to “testbed/dir2”). Such questions require that certain directories should exist (otherwise, task completion would be trivial). With that said, varying the file system dynamically could be a future work that makes for a more challenging setting than static layouts.
> - Compiled languages like C++: see **General Response 2**.
>
>
> ### 3. Vicuna and Starchat tested on SQL bench
> Vicuna and Starchat experiments on SQL are in Appendix B.2. These results were completed by the supplementary deadline. We will move these results to Table 2 in the main paper.

---

> ### Author Response · Authors · 2023-08-25
> **Look forward to discussion**
>
> Thanks again for your time and feedback!
>
> As the rebuttal is ending soon, we'd like to see if your concerns have been addressed, and would be happy to engaging in further discussions if needed!

---

### Official Review · Reviewer_VtXE · 2023-07-23
**Fair setups**

**Rating:** 4
**Confidence:** 5
**Correctness:** yes
**Clarity:** yes

**Strengths:**

The work address a vital aspect in machine learning for code generation. The practice has proven that even few rounds of feedback could greatly improve the generated code quality.

**Additional Feedback:**

Fundamentally, the reviewer doesn't think setting up a docker file from existing dataset is the main obstacle for research in this area.

On the other hand, it's questionable whether the RL setting should be standardized into the benchmark. Personally, the reviewer believes it better to leave the flexibility to the model designer.

It's mentioned but not dived into that many evaluation metrics in code generation are borrowed from natural language processing, which are unsuitable for the programming languages. The contribution could be even more significant to design a better evaluation metric to accurately reflect the model performance than standardizing the interactive environment.

**Documentation:**

Yes.

**Limitations:**

As authors have discussed, as a benchmark, it contains only two environment.

**Opportunities For Improvement:**

The work should be focused on only one topic. The first half of the work is explaining the motivation and design of this standardized environment and basic RL setups. However, the experiment results grounded to 1) interactive outperforms non-interactive approaches and 2) prompting strategy makes difference.

As a benchmark suite, it currently supports only two dataset. It's far from ready to use for the boarder audience.

As a proposal of the RL methods, the experiment should show its advantage over the existing interactive coding models on the same tasks.

**Relation To Prior Work:**

Partially.

The author discussed existing datasets in different languages, and also the previous works with interactive settings. The two benefits of InterCode have been claimed

* In the same task, using InterCode could make the results comparable.

* InterCode supports multiple environments so it makes comparing across benchmarks easier.


Despite the claims, the reviewer fails to find the further explanation or experimental results to support the claims.

**Summary And Contributions:**

The work presented their effort in standardizing the RL environment for interactive programing tasks, and evaluated bash and SQL coding tasks.

---

> ### Author Response · Authors · 2023-08-18
>
> Thanks for finding our work "fair setups" and "address a vital aspect" for code generation!
>
> ### 1. Focus on only experiment results
> Thanks for finding our experiment results “make a difference”! However, as true for all papers, we believe proper motivation and formulation are critical, even though they are not as technical as the experiment part. Please refer to **General Response 3** on why we think a standardized RL environment setup is important.
>
> ### 2. Only two dataset
> Please refer to **General Response 1**, where we expanded InterCode to include Python and CTF tasks, and ran new experiments.
>
>
> ### 3. As a proposal of the RL methods, the experiment should show its advantage over the existing interactive coding models on the same tasks.
>
> As a benchmark paper, InterCode did not intend to invent new interactive coding methods. Instead, we applied some recent methods (e.g. ReAct, Plan-and-solve) on novel domains (e.g. Bash, SQL) and showed new insights.
>
>
> ### 4. Importance of Docker
>
> We agree some traditional tasks (e.g. Spider) can be run without Docker.  Yet as argued in the paper, we do believe InterCode's Docker implemenation makes the following differences:
> - **Safety**. Docker sandbox prevents dangerous code such as `rm -rf /` in bash, `os.rmdir('/')` in python, or `DELETE` in SQL from causing danger. As models are getting increasingly capable and grounded to execution environments, such a feature is vital. This safety is crucial to isolating agents' exploits applied to CTF tasks such as binary exploitation and reverse engineering which typically involve machine code level exploits.
> - **Reproducibility**. For example, some recent Python tasks have started to use third-party packages and tools. Docker can make sure the right package version is used for reproducibility.
> - **Enables new (interactive) languages and tasks**. For example, previous work could not evaluate functional correctness of Bash programs, and had to rely on surface form matches. With docker-based execution, InterCode enables executing Bash programs on environment test cases, new CTF tasks, etc.
>
> ### 5. whether the RL setting should be standardized
> Please refer to **General Response (3)**.
>
> ### 6. The contribution could be even more significant to design a better evaluation metric
> You are right, one of our contributions is enabling functional evaluation metrics based on execution, e.g. while previous Bash benchmarks had to rely on surface form matches, we enabled execution-based evaluation of functional correctness. We will better highlight this contribution in revision.

---

> ### Author Response · Authors · 2023-08-25
> **Look forward to discussion**
>
> Thanks again for your time and feedback!
>
> As the rebuttal is ending soon, we'd like to see if your concerns have been addressed, and would be happy to engaging in further discussions if needed!

---

### Official Review · Reviewer_1q7d · 2023-07-28
**The paper presents a new pragmatic programming task design, but the dataset presented in the paper has some shortcomings.**

**Rating:** 5
**Confidence:** 3
**Correctness:** Yes.
**Clarity:** Yes.

**Strengths:**

1. The paper proposed an interesting task called interactive code generation. With the improvement of LLMs' programming ability, designing such kinds of tasks may contribute to applying LLM in practical programming tasks.

2. An open-source, extensible framework was released to provide a useful interactive environment for code generation tasks in practical scenarios.

**Additional Feedback:**

No.

**Documentation:**

I think the detailed information provided in the paper is relatively sufficient.

**Ethics:**

No.

**Limitations:**

We prefer the feedback information that interacts with the environment to include guidance on gradually answering questions step by step with large language models rather than just providing debugging information as currently done.

Maybe, the CTF dataset in Figure 4 is more suitable to be an interactive code generation dataset. However,  the amount of this dataset is too small and not fully described by the paper.

**Opportunities For Improvement:**

1.  The two proposed datasets (InterCode-SQL and InterCode-Bash) only contains SQL and Bash data. However, in most cases, the interaction between the large language model and the environment on these two datasets (especially on the Bash dataset) is only contains information for code debugging, without fully demonstrating the design intention of the interactive code generation task.

2. Missing Related Work:
Missing some important published execution-based coding-related work:
-   Zhang, K., Li, Z., Li, J., Li, G., & Jin, Z. Self-Edit: Fault-Aware Code Editor for Code Generation. (ACL 2023)
-  Le, H., Wang, Y., Gotmare, A., Savarese, S., & Hoi, S.C. CodeRL: Mastering Code Generation through Pretrained Models and Deep Reinforcement Learning. (NeurIPS 2022)



**Relation To Prior Work:**

The paper should explain the relationship with the related works listed above.

**Summary And Contributions:**

The paper introduces InterCode, a framework for constructing interactive code environments with multiple types of feedback signals.
It can serve as a challenging benchmark for code generation and comprehension. The paper uses InterCode to create two interactive code environments with Bash and SQL. Based on this, the paper evaluates the existing LLMs and related agent strategies in detail.

---

> ### Author Response · Authors · 2023-08-18
>
> Thanks for finding InterCode "interesting" and "useful"!
>
> ### 1. Only contains SQL and Bash
> Please see **General Response 1 and 2**, where we expand InterCode to Python and CTF with new experiments, and discuss how to support C++/Java.
>
> ### 2. interaction ... only contains information for code debugging ... include guidance on gradually answering questions
> We focused on execution feedback (rather than human-in-the-loop feedback) because it is generic across tasks and languages, automatic and free to obtain, and shown by our experiments to greatly enhence coding performances compared with traditional seq2seq approaches, while allowing exploration of various interactive methods.
>
> We believe incorporating human feedback is a valuable future direction, and the gym-like RL environment design of InterCode will be convenient to this end (e.g. simply adding a custom `gym.ObservationWrapper` object).
>
>
> ### 3. Missing related work
> Thanks, we will cite and discuss them!
>
> ### 4. CTF dataset too small
> Thanks for finding our CTF dataset "more suitable to be an interactive code generation dataset". Indeed, CTF inherently requires multi-step interaction, which is a novel contribution.
>
>
> Please refer to **General Response 1**, where we expanded the CTF dataset from 3 to 57 instances, and ran new experiments on it. We plan to further scale it to 100+ instances soon.

---

> > ### Author Response · Authors · 2023-08-25
> > **Look forward to discussion**
> >
> > Thanks again for your time and feedback!
> >
> > As the rebuttal is ending soon, we'd like to see if your concerns have been addressed, and would be happy to engaging in further discussions if needed!

---

### Author Response · Authors · 2023-08-18
**General Response (part 1/2)**


We appreciate all reviewer’s time and feedback! We will incorporate the rebuttal content into our revision to strengthen our work.

### 1. New InterCode tasks (Python and CTF) and experiments

**During rebuttal, we have incorporated the following new languages/tasks into InterCode. They have been updated into our offical Github repo.**
* **Python**: we have added Environment + Dockerfile support for Python (Interpreter Style), and incorporated 2 Python tasks: MBPP and APPS.
* **CTF**: we only had 3 CTF task instances at submission. Now we have added refactored, ready-to-use CTF Environment + Dockerfile, which helped us scale the CTF task to 57 instances. We plan to further scale it up.
* **SQL**: Besides Spider, we also incorporated two new SQL tasks: BIRD-SQL and WikiSQL.


Based on the new tasks, we have also run the following new experiments.

__New Python Experiments__: We run Single Turn and Try Again experiments on 117 instances of the MBPP dataset in the InterCode-Python environment. The task is to write a function that implements the given MBPP `text` instruction in a Python Interpreter setting. In the Try Again setup, the agent is informed via the prompt that it can use the Python Interpreter to write, debug, and test its solution. Once it feels the implementation is correct, it specifies `submit <function name>` as the action. The agent is given a maximum of 7 turns.

We run experiments on GPT 3.5 and GPT 4. All material (i.e. full prompts, evaluation scripts, code environments) have already been uploaded for reproducibility.

_Experiment 1_: GPT 3.5 evaluation on MBPP dataset in InterCode Python (117 instances)
* Multi Turn Avg. Reward: 0.598
* Single Turn Avg. Reward: 0.469

|                            |    |    |    |    |    |    |    |
|----------------------------|----|----|----|----|----|----|----|
| __Turn__                   |  1 |  2 |  3 |  4 |  5 |  6 |  7 |
| __# Tasks Solved by Turn__ | 51 | 64 | 68 | 68 | 69 | 69 | 69 |

_Experiment 2_: GPT 4 evaluation on MBPP dataset in InterCode Python (117 instances)
* Multi Turn Avg. Reward: 0.635
* Single Turn Avg. Reward: 0.538

|                            |    |    |    |    |    |    |    |
|----------------------------|----|----|----|----|----|----|----|
| __Turn__                   |  1 |  2 |  3 |  4 |  5 |  6 |  7 |
| __# Tasks Solved by Turn__ | 61 | 67 | 68 | 68 | 69 | 69 | 70 |

__New CTF Experiments__: We run Try Again experiments on 50 instances of the forthcoming CTF dataset in the InterCode-CTF environment. The task is to find and submit a flag given an instruction and task related resources in an Ubuntu OS setting. In the Try Again setup, the agent is informed via the prompt that it can use Bash commands, use Python commands, or write + execute self-generated Python scripts. Once the agent feels it has found the correct flag, it specifies `submit <flag>` as the action. The agent is given a maximum of 10 turns.

We run experiments on GPT 4 (constraining GPT 3.5 generations with prompts to interact reliably with this environment proved too hard). All materials (i.e. full prompts, evaluation scripts, code environments) have been uploaded for reproducibility.

_Experiment 3_: GPT 4 evaluation on CTF dataset in InterCode CTF (50 instances)
* Multi Turn Avg. Reward: 0.510
* Single Turn Avg. Reward: 0

|                            |    |    |    |    |    |    |    |    |    |    |
|----------------------------|----|----|----|----|----|----|----|----|----|----|
| __Turn__                   |  1 |  2 |  3 |  4 |  5 |  6 |  7 |  8 |  9 | 10 |
| __# Tasks Solved by Turn__ |  0 |  7 | 12 | 18 | 21 | 24 | 25 | 26 | 26 | 26 |

### 2. How to support compiled languages (i.e. Java, C++)

We see two viable avenues of support for compiled languages (i.e. C, C++, Java, Go, Rust):

- 3rd party interpreter support (Less preferable): Like the Python interpreter, tools such as JShell (for Java) or Yaegi (for Go) may be serviceable interpreters for enabling REPL style code interaction. However, this usage style is a bit contrived and not really found in real world software development processes.
- Bash + (Compiled Language): Make a compiler (i.e. gcc, javac) available in an InterCode-Bash style environment. This allows an agent to use Bash commands to create, write to and execute compiled-language files. This is possible with InterCode-CTF.

As a side note, NL2Code datasets based on C/C++/Java/etc have been rare. By designing a language-agnostic framework for coding tasks, we hope InterCode encourages more exploration into such tasks.

(part 1/2)

---

> ### Author Response · Authors · 2023-08-18
> **General Response (part 2/2)**
>
> ### 3. InterCode’s contribution as a standardized framework
>
> Our paper's contribution is twofold:
> - conceptually, it proposes to formalize various coding tasks (intrinsically interactive or not) as a standard RL interaction problem, with action being code and observation being execution feedback.
> - empirically, it proposes a multi-task coding benchmark and associated Environment+Dockfile supports, allows execution-based methods and evaluations, and presents various experiment findings.
>
> While the value of the empirical contributions is agreed by all reviewers, we believe the conceptual contribution is also important for the following reasons:
> - the RL setup is simple yet generic. it naturally includes the traditional seq2seq task or model setups (as a bandit special case), while more importantly, allows exploration of new language and tasks (e.g. the inherently multi-language and multi-step CTF), evaluations (e.g. Bash execution), and methods (e.g. ReAct, Plan-then-Solve).
> - while there has been some interactive/execution-based coding tasks/methods proposed recently, the method is often entangled with a customly written task-based execution interaction, often making methods task-specific, and making different methods hard to compare due to different implementations of interaction/execution. By neatly disentangling the agent and environment, the RL setup solves the above problems, and makes it modular and easy to write new tasks or methods without worrying about the other part.
>
> Analogous to the value OpenAI Gym has brought to the field of RL, we believe InterCode's clean abstraction and Github-accessible implementation will bring value to the increasingly interactive and execution-based field of coding.
>
> We also encourage reviewers to check why we focused on execution feedback (**1q7d#2**), the value of Docker-based implementation (**VtXE#4**), and the contribution of InterCode's execution-based evaluation metrics (**VtXE#6**).
>
> (part 2/2)

---

### Decision · Program_Chairs · 2023-09-22

**Decision:**

Accept (Poster)

**Comment:**

The paper introduces a coding framework that supports execution feedback. An agent can interact with the InterCode environment by executing the generated response, receiving feedback, and refining the answer iteratively. While the framework is extensible to various tasks and programming language, the authors propose in the main paper two benchmarks on two interactive code tasks with Bash and SQL and evaluate multiple SoTA LLM. The results show that with multi-turn prompting and interactive feedback, models exhibit the capability of error correction and context discovery, which significantly improves their performance compared to the single-turn baseline.

All the reviewers agree on the importance of the problem tackled, the novelty of the approach but were doubtful about the exhaustiveness of the study. The authors provided in the rebuttal additional tasks and programming language and discussed the generalization of the approach to broader tasks and to compiled languages. The tool is provided in a Docker format that is both ready to use and extensible and I believe the framework and benchmark opens new venues of improvement of LLM for code related tasks.